# The Complexity of Adversarially Robust Proper Learning of Halfspaces with Agnostic Noise

**Ilias Diakonikolas**
University of Wisconsin, Madison
ilias@cs.wisc.edu

**Daniel M. Kane**
University of California, San Diego
dakane@cs.ucsd.edu

**Pasin Manurangsi**
Google Research, Mountain View
pasin@google.com

## Abstract

We study the computational complexity of adversarially robust proper learning of halfspaces in the distribution-independent agnostic PAC model, with a focus on $L_p$ perturbations. We give a computationally efficient learning algorithm and a nearly matching computational hardness result for this problem. An interesting implication of our findings is that the $L_\infty$ perturbations case is provably computationally harder than the case $2 \le p < \infty$.

## 1 Introduction

In recent years, the design of reliable machine learning systems for secure-critical applications, including in computer vision and natural language processing, has been a major goal in the field. One of the main concrete goals in this context has been to develop classifiers that are robust to *adversarial examples*, i.e., small imperceptible perturbations to the input that can result in erroneous misclassification [BCM+13, SZS+14, GSS15]. This has led to an explosion of research on designing defenses against adversarial examples and attacks on these defenses. See, e.g., [KM18] for a recent tutorial on the topic. Despite significant empirical progress over the past few years, the broad question of designing computationally efficient classifiers that are provably robust to adversarial perturbations remains an outstanding theoretical challenge.

In this paper, we focus on understanding the *computational complexity* of adversarially robust classification in the (distribution-independent) agnostic PAC model [Hau92, KSS94]. Specifically, we study the learnability of *halfspaces* (or linear threshold functions) in this model with respect to $L_p$ perturbations. A halfspace is any function $h_{\mathbf{w}} : \mathbb{R}^d \to \{\pm 1\}$ of the form[1] $h_{\mathbf{w}}(\mathbf{x}) = \mathrm{sgn}\left(\langle \mathbf{w}, \mathbf{x} \rangle\right)$, where $\mathbf{w} \in \mathbb{R}^d$ is the associated weight vector. The problem of learning an unknown halfspace has been studied for decades — starting with the Perceptron algorithm [Ros58] — and has arguably been one of the most influential problems in the development of machine learning [Vap98, FS97].

Before we proceed, we introduce the relevant terminology. Let $\mathcal{C}$ be a concept class of Boolean-valued functions on an instance space $\mathcal{X} \subseteq \mathbb{R}^d$ and $\mathcal{H}$ be a hypothesis class on $\mathcal{X}$. The set of allowable perturbations is defined by a function $\mathcal{U} : \mathcal{X} \to 2^{\mathcal{X}}$. The robust risk of a hypothesis $h \in \mathcal{H}$ with respect to a distribution $\mathcal{D}$ on $\mathcal{X} \times \{\pm 1\}$ is defined as $\mathcal{R}_{\mathcal{U}}(h, \mathcal{D}) = \Pr_{(\mathbf{x},y) \sim \mathcal{D}}[\exists z \in \mathcal{U}(\mathbf{x}), h(\mathbf{z}) \ne y]$. The (adversarially robust) agnostic PAC learning problem for $\mathcal{C}$ is the following: Given i.i.d. samples from an arbitrary distribution $\mathcal{D}$ on $\mathcal{X} \times \{\pm 1\}$, the goal of the learner is to output a hypothesis $h \in \mathcal{H}$

such that with high probability it holds $\mathcal{R}_{\mathcal{U}}(h, \mathcal{D}) \leq \text{OPT}^{\mathcal{D}} + \epsilon$, where $\text{OPT}^{\mathcal{D}} = \inf_{f \in \mathcal{C}} \mathcal{R}_{\mathcal{U}}(f, \mathcal{D})$ is the robust risk of the best-fitting function in $\mathcal{C}$.

Unfortunately, it follows from known hardness results that this formulation is computationally intractable for the class of halfspaces $\mathcal{C} = \{\text{sgn}(\langle \mathbf{w}, \mathbf{x} \rangle), \mathbf{w} \in \mathbb{R}^d\}$ under $L_p$ perturbations, i.e, for $\mathcal{U}_{p,\gamma}(\mathbf{x}) = \{\mathbf{z} \in \mathcal{X} : \|\mathbf{z} - \mathbf{x}\|_p \leq \gamma\}$, for some $p \geq 2$. (The reader is referred to the supplementary material for a more detailed explanation.) To be able to obtain computationally efficient algorithms, we relax the above definition in two ways: (1) We allow the hypothesis to be robust within a slightly smaller perturbation region, and (2) We introduce a small constant factor approximation in the error guarantee. In more detail, for some constants $0 < \nu < 1$ and $\alpha > 1$, our goal is to efficiently compute a hypothesis $h$ such that with high probability

$$\mathcal{R}_{\mathcal{U}_{p,(1-\nu)\gamma}}(h, \mathcal{D}) \leq \alpha \cdot \text{OPT}^{\mathcal{D}}_{p,\gamma} + \epsilon \,, \tag{1}$$

where $\text{OPT}^{\mathcal{D}}_{p,\gamma} = \inf_{f \in \mathcal{C}} \mathcal{R}_{\mathcal{U}_{p,\gamma}}(f, \mathcal{D})$. (Note that for $\nu = 0$ and $\alpha = 1$, we obtain the original definition.) An interesting setting is when $\nu$ is a small constant close to 0, say $\nu = 0.1$, and $\alpha = 1 + \delta$, where $0 < \delta < 1$. In this paper, we characterize the computational complexity of this problem with respect to *proper* learning algorithms, i.e., algorithms that output a halfspace hypothesis.

Throughout this paper, we will assume that the domain of our functions is bounded in the $d$-dimensional $L_p$ unit ball $\mathbb{B}^d_p$. All our results immediately extend to general domains with a (necessary) dependence on the diameter of the feasible set.

A simple but crucial observation leveraged in our work is the following: The adversarially robust learning problem of halfspaces under $L_p$ perturbations (defined above) is essentially equivalent to the classical problem of agnostic proper PAC learning of halfspaces with an $L_p$ margin.

Let $p \geq 2$, $q$ be the dual exponent of $p$, i.e., $1/p + 1/q = 1$. The problem of agnostic proper PAC learning of halfspaces with an $L_p$ margin is the following: The learner is given i.i.d. samples from a distribution $\mathcal{D}$ over $\mathbb{B}^d_p \times \{\pm 1\}$. For $\mathbf{w} \in \mathbb{B}^d_q$, its $\gamma$-*margin error* is defined as $\text{err}^{\mathcal{D}}_\gamma(\mathbf{w}) := \Pr_{(\mathbf{x},y) \sim \mathcal{D}}[\text{sgn}(\langle \mathbf{w}, \mathbf{x} \rangle - y \cdot \gamma) \neq y]$. We also define $\text{OPT}^{\mathcal{D}}_\gamma := \min_{\mathbf{w} \in \mathbb{B}^d_q} \text{err}^{\mathcal{D}}_\gamma(\mathbf{w})$. An algorithm is a proper $\nu$-*robust* $\alpha$-*agnostic learner* for $L_p$-$\gamma$-margin halfspace if, with probability at least $1 - \tau$, it outputs a halfspace $\mathbf{w} \in \mathbb{B}^d_q$ with

$$\text{err}^{\mathcal{D}}_{(1-\nu)\gamma}(\mathbf{w}) \leq \alpha \cdot \text{OPT}^{\mathcal{D}}_\gamma + \epsilon \,. \tag{2}$$

(When unspecified, the failure probability $\tau$ is assumed to be 1/3. It is well-known and easy to see that we can always achieve arbitrarily small value of $\tau$ at the cost of $O(\log(1/\tau))$ multiplicative factor in the running time and sample complexity.)

We have the following basic observation, which implies that the learning objectives (1) and (2) are equivalent. Throughout this paper, we will state our contributions using the margin formulation (2).

**Fact 1.** *For any non-zero* $\mathbf{w} \in \mathbb{R}^d$, $\gamma \geq 0$ *and* $\mathcal{D}$ *over* $\mathbb{R}^d \times \{\pm 1\}$, $\mathcal{R}_{\mathcal{U}_{p,\gamma}}(h_{\mathbf{w}}, \mathcal{D}) = \text{err}^{\mathcal{D}}_\gamma(\frac{\mathbf{w}}{\|\mathbf{w}\|_q})$.

## 1.1 Our Contributions

Our main positive result is a robust and agnostic proper learning algorithm for $L_p$-$\gamma$-margin halfspace with near-optimal running time:

**Theorem 2** (Robust Learning Algorithm). *Fix* $2 \leq p < \infty$ *and* $0 < \gamma < 1$. *For any* $0 < \nu, \delta < 1$, *there is a proper* $\nu$-*robust* $(1 + \delta)$-*agnostic learner for* $L_p$-$\gamma$-*margin halfspace that draws* $O(\frac{p}{\epsilon^2 \nu^2 \gamma^2})$ *samples and runs in time* $(1/\delta)^{O\left(\frac{p}{\nu^2 \gamma^2}\right)} \cdot \text{poly}(d/\epsilon)$.

*Furthermore, for* $p = \infty$, *there is a proper* $\nu$-*robust* $(1 + \delta)$-*agnostic learner for* $L_\infty$-$\gamma$-*margin halfspace that draws* $O(\frac{\log d}{\epsilon^2 \nu^2 \gamma^2})$ *samples and runs in time* $d^{O\left(\frac{\log(1/\delta)}{\nu^2 \gamma^2}\right)} \cdot \text{poly}(1/\epsilon)$.

To interpret the running time of our algorithm, we consider the setting $\delta = \nu = 0.1$. We note two different regimes. If $p \geq 2$ is a fixed constant, then our algorithm runs in time $2^{O(1/\gamma^2)} \text{poly}(d/\epsilon)$. On the other hand, for $p = \infty$, we obtain a runtime of $d^{O(1/\gamma^2)} \text{poly}(1/\epsilon)$. That is, the $L_\infty$ margin case (which corresponds to adversarial learning with $L_\infty$ perturbations) appears to be computationally the hardest. As we show in Theorem 3, this fact is inherent for proper learners.

Our algorithm establishing Theorem 2 follows via a simple and unified approach, employing a reduction from online (mistake bound) learning [Lit87]. Specifically, we show that any computationally efficient $L_p$ online learner for halfspaces with margin guarantees and mistake bound $M$ can be used in a black-box manner to obtain an algorithm for our problem with runtime roughly $\mathrm{poly}(d/\epsilon)(1/\delta)^M$. Theorem 2 then follows by applying known results from the online learning literature [Gen01a].

For the special case of $p = 2$ (and $\nu = 0.1$), recent work [DKM19] gave a sophisticated algorithm for our problem with running time $\mathrm{poly}(d/\epsilon)2^{\tilde{O}(1/(\delta\gamma^2))}$. We note that our algorithm has significantly better dependence on the parameter $\delta$ (quantifying the approximation ratio), and better dependence on $1/\gamma$. Importantly, our algorithm is much simpler and immediately generalizes to all $L_p$ norms.

Perhaps surprisingly, the running time of our algorithm is nearly the best possible for proper learning. For constant $p \geq 2$, this follows from the hardness result of [DKM19]. (See the supplementary material for more details.) Furthermore, we prove a tight running time lower bound for robust $L_\infty$-$\gamma$-margin proper learning of halfspaces. Roughly speaking, we show that for some sufficiently small constant $\nu > 0$, one cannot hope to significantly speed-up our algorithm for $\nu$-robust $L_\infty$-$\gamma$-margin learning of halfspaces. Our computational hardness result is formally stated below.

**Theorem 3** (Tight Running Time Lower Bound). *There exists a constant $\nu > 0$ such that, assuming the (randomized) Gap Exponential Time Hypothesis, there is no proper $\nu$-robust 1.5-agnostic learner for $L_\infty$-$\gamma$-margin halfspace that runs in time $f(1/\gamma) \cdot d^{o(1/\gamma^2)} \mathrm{poly}(1/\epsilon)$ for any function $f$.*

As indicated above, our running time lower bound is based on the so-called Gap Exponential Time Hypothesis (Gap-ETH), which roughly states that no subexponential time algorithm can approximate 3SAT to within $(1 - \epsilon)$ factor, for some constant $\epsilon > 0$. Since we will not be dealing with Gap-ETH directly here, we defer the formal treatment of the hypothesis and discussions on its application to the supplementary material.

We remark that the constant 1.5 in our theorem is insignificant. We can increase this "gap" to any constant less than 2. We use the value 1.5 to avoid introducing an additional variable. Another remark is that Theorem 3 only applies for a small constant $\nu > 0$. This leaves the possibility of achieving, e.g., a faster 0.9-robust $L_\infty$-$\gamma$-margin learner for halfspaces, as an interesting open problem.

## 1.2 Related Work

A sequence of recent works [CBM18, SST+18, BLPR19, MHS19] has studied the sample complexity of adversarially robust PAC learning for general concept classes of bounded VC dimension and for halfspaces in particular. [MHS19] established an upper bound on the sample complexity of PAC learning any concept class with finite VC dimension. A common implication of the aforementioned works is that, for some concept classes, the sample complexity of adversarially robust PAC learning is higher than the sample complexity of (standard) PAC learning. For the class of halfspaces, which is the focus of the current paper, the sample complexity of adversarially robust agnostic PAC learning was shown to be essentially the same as that of (standard) agnostic PAC learning [CBM18, MHS19].

Turning to computational aspects, [BLPR19, DNV19] showed that there exist classification tasks that are efficiently learnable in the standard PAC model, but are computationally hard in the adversarially robust setting (under cryptographic assumptions). Notably, the classification problems shown hard are artificial, in the sense that they do not correspond to natural concept classes. [ADV19] shows that adversarially robust proper learning of degree-2 polynomial threshold functions is computationally hard, even in the realizable setting. On the positive side, [ADV19] gives a polynomial-time algorithm for adversarially robust learning of halfspaces under $L_\infty$ perturbations, again in the realizable setting. More recently, [MGDS20] generalized this upper bound to a broad class of perturbations, including $L_p$ perturbations. Moreover, [MGDS20] gave an efficient algorithm for learning halfspaces with random classification noise [AL88]. We note that all these algorithms are proper.

The problem of agnostically learning halfspaces with a margin has been studied extensively. A number of prior works [BS00, SSS09, SSS10, LS11, BS12, DKM19] studied the case of $L_2$ margin and gave a range of time-accuracy tradeoffs for the problem. The most closely related prior work is the recent work [DKM19], which gave a proper $\nu$-robust $\alpha$-agnostic learning for $L_2$-$\gamma$-margin halfspace with near-optimal running time when $\alpha, \nu$ are universal constants, and a nearly matching computational hardness result. The algorithm of the current paper broadly generalizes, simplifies, and improves the algorithm of [DKM19].

## 2 Upper Bound: From Online to Adversarially Robust Agnostic Learning

In this section, we provide a generic method that turns an online (mistake bound) learning algorithm for halfspaces into an adversarially robust agnostic algorithm, which is then used to prove Theorem 2.

Recall that online learning [Lit87] proceeds in a sequence of rounds. In each round, the algorithm is given an example point, produces a binary prediction on this point, and receives feedback on its prediction (after which it is allowed to update its hypothesis). The mistake bound of an online learner is the maximum number of mistakes (i.e., incorrect predictions) it can make over all possible sequences of examples.

We start by defining the notion of online learning with a margin gap in the context of halfspaces:

**Definition 4.** *An online learner $\mathcal{A}$ for the class of halfspaces is called an $L_p$ online learner with mistake bound $M$ and $(\gamma, \gamma')$ margin gap if it satisfies the following: In each round, $\mathcal{A}$ returns a vector $\mathbf{w} \in \mathbb{B}_q^d$. Moreover, for any sequence of labeled examples $(\mathbf{x}_i, y_i)$ such that there exists $\mathbf{w}^* \in \mathbb{B}_q^d$ with $\mathrm{sgn}(\langle \mathbf{w}^*, \mathbf{x}_i \rangle - y_i \gamma) = y_i$ for all $i$, there are at most $M$ values of $t$ such that $\mathrm{sgn}(\langle \mathbf{w}_t, \mathbf{x}_t \rangle - y_t \gamma') \neq y_t$, where $\mathbf{w}_t = \mathcal{A}((\mathbf{x}_1, y_1), \ldots, (\mathbf{x}_{t-1}, y_{t-1}))$.*

The $L_p$ online learning problem of halfspaces has been studied extensively in the literature, see, e.g., [Lit87, GLS01, Gen01b, Gen03, BB14]. We will use a result of [Gen01a], which gives a polynomial time $L_p$ online learner with margin gap $(\gamma, (1-\nu)\gamma)$ and mistake bound $O((p-1)/\nu^2\gamma^2)$.

We are now ready to state our generic proposition that translates an online algorithm with a given mistake bound into an agnostic learning algorithm. We will use the following notation: For $S \subseteq \mathbb{B}_p^d \times \{\pm 1\}$, we will use $S$ instead of $\mathcal{D}$ to denote the empirical error on the uniform distribution over $S$. In particular, we denote $\mathrm{err}_\gamma^S(\mathbf{w}) := \frac{1}{|S|} \cdot |\{(\mathbf{x}, y) \in S \mid \mathrm{sgn}(\langle \mathbf{w}, \mathbf{x} \rangle - y\gamma) \neq y\}|$.

The main result of this section is the following proposition. While we state our proposition for the empirical error, it is simple to convert it into a generalization bound as we will show later in the proof of Theorem 2.

**Proposition 5.** *Assume that there is a polynomial time $L_p$ online learner $\mathcal{A}$ for halfspaces with a $(\gamma, \gamma')$ margin gap and mistake bound of $M$. Then there exists an algorithm that given a multiset of labeled examples $S \subseteq \mathbb{B}_p^d \times \{\pm 1\}$ and $\delta \in (0, 1)$, runs in $\mathrm{poly}(|S|d) \cdot 2^{O(M \log(1/\delta))}$ time and with probability $9/10$ returns $\mathbf{w} \in \mathbb{B}_q^d$ such that $\mathrm{err}_{\gamma'}^S(\mathbf{w}) \leq (1 + \delta) \cdot \mathrm{OPT}_\gamma^S$.*

Notice that our algorithm runs in time $\mathrm{poly}(|S|d) \cdot 2^{O(M \log(1/\delta))}$ and has success probability $9/10$. It is more convenient to describe a version of our algorithm that runs in $\mathrm{poly}(|S|d)$ time, but has small success probability of $2^{-O(M \log(1/\delta))}$, as encapsulated by the following lemma.

**Lemma 6.** *Assume that there is a polynomial time $L_p$ online learner $\mathcal{A}$ for halfspaces with a $(\gamma, \gamma')$ margin gap and mistake bound of $M$. Then there exists an algorithm that given a multiset of labeled examples $S \subseteq \mathbb{B}_p^d \times \{\pm 1\}$ and $\delta \in (0, 1)$, runs in $\mathrm{poly}(|S|dM)$ time and with probability $2^{-O(M \log(1/\delta))}$ returns $\mathbf{w} \in \mathbb{B}_q^d$ such that $\mathrm{err}_{\gamma'}^S(\mathbf{w}) \leq (1 + \delta) \cdot \mathrm{OPT}_\gamma^S$.*

Before proving Lemma 6, notice that Proposition 5 now follows by running the algorithm from Lemma 6 independently $2^{O(M \log(1/\delta))}$ times and returning the $\mathbf{w}$ with minimum $\mathrm{err}_{\gamma'}^S(\mathbf{w})$. Since each iteration has a $2^{-O(M \log(1/\delta))}$ probability of returning a $\mathbf{w}$ with $\mathrm{err}_{\gamma'}^S(\mathbf{w}) \leq (1 + \delta) \cdot \mathrm{OPT}_\gamma^S$, with 90% probability at least one of our runs finds a $\mathbf{w}$ that satisfies this.

*Proof of Lemma 6.* Let $\mathbf{w}^* \in \mathbb{B}_q^d$ denote an "optimal" halfspace with $\mathrm{err}_\gamma^S(\mathbf{w}^*) = \mathrm{OPT}_\gamma^S$.

The basic idea of the algorithm is to repeatedly run $\mathcal{A}$ on larger and larger subsets of samples each time adding one additional sample in $S$ that the current hypothesis gets wrong. The one worry here is that some of the points in $S$ might be errors, inconsistent with the true classifier $\mathbf{w}^*$, and feeding them to our online learner will lead it astray. However, at any point in time, either we misclassify (w.r.t. margin $\gamma'$) only a $(1 + \delta) \cdot \mathrm{OPT}_\gamma^S$ fraction of points (in which case we can abort early and use this hypothesis) or guessing a random misclassified point will have at least an $\Omega(\delta)$ probability of giving us a non-error. Since our online learner has a mistake bound of $M$, we will never need to make more than this many correct guesses. Specifically, the algorithm is as follows:

- Let Samples $= \emptyset$
- For $i = 0$ to $M$
    - Let $\mathbf{w} = \mathcal{A}(\text{Samples})$
    - Let $T$ be the set of $(\mathbf{x}, y) \in S$ so that $\text{sgn}(\langle \mathbf{w}, \mathbf{x} \rangle - y\gamma') \neq y$
    - If $T = \emptyset$, and otherwise with $50\%$ probability, return $\mathbf{w}$
    - Draw $(\mathbf{x}_i, y_i)$ uniformly at random from $T$, and add it to Samples
- Return $\mathbf{w}$

To analyze this algorithm, let $S_{bad}$ be the set of $(\mathbf{x}, y) \in S$ with $\text{sgn}(\langle \mathbf{w}^*, \mathbf{x} \rangle - y\gamma) \neq y$. Recall that by assumption $|S_{bad}| \leq \text{OPT}_\gamma^S \cdot |S|$. We claim that with probability at least $2^{-O(M \log(1/\delta))}$ our algorithm never adds an element of $S_{bad}$ to Samples and never returns a $\mathbf{w}$ in the for loop for which $\text{err}_{\gamma'}^S(\mathbf{w}) > (1 + \delta) \cdot \text{OPT}_\gamma^S$. This is because during each iteration of the algorithm either:

1. $\text{err}_{\gamma'}^S(\mathbf{w}) > (1 + \delta) \cdot \text{OPT}_\gamma^S$. In this case, there is a $50\%$ probability that we do not return $\mathbf{w}$. If we do not return, then $|T| \geq (1 + \delta) \cdot |S_{bad}|$ so there is at least a $\frac{\delta}{1+\delta} \geq \delta/2$ probability that the new element added to Samples is not in $S_{bad}$.

2. Or $\text{err}_{\gamma'}^S(\mathbf{w}) \leq (1 + \delta) \cdot \text{OPT}_\gamma^S$. In this case, there is a $50\%$ probability of returning $\mathbf{w}$.

Hence, there is a $(\delta/4)^{M+1} \geq 2^{-O(M \log(1/\delta))}$ probability of never adding an element of $S_{bad}$ to Samples or returning a $\mathbf{w}$ in our for-loop with $\text{err}_{\gamma'}^S(\mathbf{w}) > (1+\delta) \cdot \text{OPT}_\gamma^S$. When this occurs, we claim that we output $\mathbf{w}$ such that $\text{err}_{\gamma'}^S(\mathbf{w}) \leq (1 + \delta) \cdot \text{OPT}_\gamma^S$. This is because, if this were not the case, we must have reached the final statement at which point we have Samples $= ((\mathbf{x}_0, y_0), \ldots, (\mathbf{x}_M, y_M))$, where each $(\mathbf{x}_i, y_i)$ satisfies $\text{sgn}(\langle \mathbf{w}^*, \mathbf{x}_i \rangle - y_i\gamma) = y_i$ and $\text{sgn}(\langle \mathbf{w}_i, \mathbf{x}_i \rangle - y_i\gamma') \neq y_i$ with $\mathbf{w}_i = \mathcal{A}((\mathbf{x}_0, y_0), \ldots, (\mathbf{x}_{i-1}, y_{i-1}))$. But this violates the mistake bound of $M$.

Thus, we output $\mathbf{w}$ such that $\text{err}_{\gamma'}^S(\mathbf{w}) \leq (1+\delta) \cdot \text{OPT}_\gamma^S$ with probability at least $2^{-O(M \log(1/\delta))}$. $\quad\square$

We will now show how Proposition 5 can be used to derive Theorem 2. As stated earlier, we will require the following mistake bound for online learning with a margin gap from [Gen01a].

**Theorem 7** ([Gen01a]). *For any $2 \leq p < \infty$, there exists a polynomial time $L_p$ online learner with margin gap $(\gamma, (1 - \nu)\gamma)$ and mistake bound $O\left(\frac{(p-1)}{\nu^2\gamma^2}\right)$. Furthermore, there is a polynomial time $L_\infty$ online learner with margin gap $(\gamma, (1 - \nu)\gamma)$ and mistake bound $O\left(\frac{\log d}{\nu^2\gamma^2}\right)$.*

*Proof of Theorem 2.* Our $\nu$-robust $(1 + \delta)$-agnostic learner for $L_p$-$\gamma$-margin halfspace works as follows. First, it draws the appropriate number of samples $m$ (as stated in Theorem 2) from $\mathcal{D}$. Then, it runs the algorithm from Proposition 5 on these samples for margin gap $(\gamma, (1 - \nu/2)\gamma)$.

Let $M_p$ denote the error bound for $L_p$ online learning with margin gap $(\gamma, (1 - \nu/2)\gamma)$ given by Theorem 7. Our entire algorithm runs in time $\text{poly}(m) \cdot 2^{O(M_p \cdot \log(1/\delta))}$. It is simple to check that this results in the claimed running time.

As for the error guarantee, let $\mathbf{w} \in \mathbb{B}_q^d$ be the output halfspace. With probability 0.8, we have

$$\text{err}_{(1-\nu)\gamma}^{\mathcal{D}}(\mathbf{w}) \leq \text{err}_{(1-\nu/2)\gamma}^S(\mathbf{w}) + \epsilon/2 \leq (1 + \delta) \cdot \text{OPT}_{(1-\nu/2)\gamma}^S + \epsilon/2 \leq (1 + \delta) \cdot \text{OPT}_\gamma^{\mathcal{D}} + \epsilon,$$

where the first and last inequalities follow from standard margin generalization bounds [BM02, KP02, KST08] and the second inequality follows from the guarantee of Proposition 5. $\quad\square$

## 3 Tight Running Time Lower Bound: Proof Overview

We will now give a high-level overview of our running time lower bound (Theorem 3). Due to space constraint, we will sometimes be informal; everything will be formalized in the supplementary material.

The main component of our hardness result will be a reduction from the *Label Cover* problem[2], which is a classical problem in hardness of approximation literature that is widely used as a starting point for proving strong NP-hardness of approximation results (see, e.g., [ABSS97, Hås96, Hås01, Fei98]).

**Definition 8** (Label Cover). *A Label Cover instance* $\mathcal{L} = (U, V, E, \Sigma_U, \Sigma_V, \{\pi_e\}_{e\in\Sigma})$ *consists of*

- *a bi-regular bipartite graph* $(U, V, E)$, *referred to as the* constraint graph,

- label sets $\Sigma_U$ *and* $\Sigma_V$,

- *for every edge* $e \in E$, *a* constraint *(aka* projection*)* $\pi_e : \Sigma_U \to \Sigma_V$.

*A labeling of* $\mathcal{L}$ *is a function* $\phi : U \to \Sigma_U$. *We say that* $\phi$ *covers* $v \in V$ *if there exists* $\sigma_v \in \Sigma_V$ *such that*[3] $\pi_{(u,v)}(\phi(u)) = \sigma_v$ *for all*[4] $u \in N(v)$. *The* value $\phi$, *denoted by* $\mathrm{val}_{\mathcal{L}}(\phi)$, *is defined as the fraction of* $v \in V$ *covered by* $\phi$. *The value of* $\mathcal{L}$, *denoted by* $\mathrm{val}(\mathcal{L})$, *is defined as* $\max_{\phi:U\to\Sigma_U} \mathrm{val}(\phi)$.

*Moreover, we say that* $\phi$ *weakly covers* $v \in V$ *if there exist distinct neighbors* $u_1, u_2$ *of* $v$ *such that* $\pi_{(u_1,v)}(\phi(u_1)) = \pi_{(u_2,v)}(\phi(u_2))$. *The* weak value *of* $\phi$, *denoted by* $\mathrm{wval}(\phi)$, *is the fraction of* $v \in V$ *weakly covered by* $\phi$. *The weak value of* $\mathcal{L}$, *denoted by* $\mathrm{wval}(\mathcal{L})$, *is defined as* $\max_{\phi:U\to\Sigma_U} \mathrm{wval}(\phi)$.

*For a Label Cover instance* $\mathcal{L}$, *we use* $k$ *to denote* $|U|$ *and* $n$ *to denote* $|U| \cdot |\Sigma_U| + |V| \cdot |\Sigma_V|$.

The goal of Label Cover is to find an assignment with maximum value. Several strong inapproximability results for Label Cover are known [Raz98, MR10, DS14]. To prove a tight running time lower bound, we require an inapproximability result for Label Cover with a tight running lower bound as well. Observe that we can solve Label Cover in time $n^{O(k)}$ by enumerating through all possible assignments and compute their values. The following result shows that, even if we aim for a constant approximation ratio, no algorithm that can be significantly faster than this "brute-force" algorithm.

**Theorem 9** ([Man20]). *Assuming Gap-ETH, for any function* $f$ *and any constant* $\mu \in (0, 1)$, *no* $f(k) \cdot n^{o(k)}$*-time algorithm can, given a Label Cover instance* $\mathcal{L}$, *distinguish between the following two cases: (Completeness)* $\mathrm{val}(\mathcal{L}) = 1$, *and, (Soundness)* $\mathrm{wval}(\mathcal{L}) < \mu$.

Given a Label Cover instance $\mathcal{L}$, our reduction produces an oracle $\mathcal{O}$ that can sample (in polynomial time) from a distribution $\mathcal{D}$ over $\mathbb{B}_\infty^d \times \{\pm 1\}$ (for some $d \leq n$) such that:

- (Completeness) If $\mathrm{val}(\mathcal{L}) = 1$, then $\mathrm{OPT}_{\gamma^*}^{\mathcal{D}} \leq \epsilon^*$.

- (Soundness) If $\mathrm{wval}(\mathcal{L}) < \mu$, then $\mathrm{OPT}_{(1-\nu)\gamma^*}^{\mathcal{D}} > 1.6\epsilon^*$.

- (Margin and Error Bounds) $\gamma^* = \Omega(1/\sqrt{k})$ and $\epsilon^* = 1/n^{o(k)}$.

Here $\nu > 0$ is some constant. Once we have such a reduction, Theorem 3 follows quite easily. The reason is that, if we assume (by contrapositive) that there exists a $\nu$-robust 1.5-agnostic learner $\mathcal{A}$ for $L_\infty$-$\gamma$-margin halfspaces that runs in time $f(1/\gamma) \cdot d^{o(1/\gamma^2)} \mathrm{poly}(1/\epsilon)$, then we can turn $\mathcal{A}$ to an algorithm for Label Cover by first using the reduction above to give us an oracle $\mathcal{O}$ and then running $\mathcal{A}$ on $\mathcal{O}$. With appropriate parameters, $\mathcal{A}$ can distinguish between the two cases in Theorem 9 in time $f(1/\gamma^*) \cdot d^{o(1/(\gamma^*)^2)} \mathrm{poly}(1/\epsilon^*) = f(O(\sqrt{k})) \cdot n^{o(k)}$, which by Theorem 9 violates the randomized Gap-ETH. Therefore, we will henceforth focus on the reduction and its proof of correctness.

**Previous Results.** To explain the key new ideas behind our reduction, it is important to understand high-level approaches taken in previous works and why they fail to yield running time lower bounds as in our Theorem 3.

Most of the known hardness results for agnostic learning of halfspaces employ reductions from Label Cover [ABSS97, FGKP06, GR09, FGRW12, DKM19][5]. These reductions use gadgets which are "local" in nature. As we will explain next, such "local" reductions *cannot* work for our purpose.

To describe the reductions, it is convenient to think of each sample $(\mathbf{x}, y)$ as a linear constraint $\langle \mathbf{w}, \mathbf{x} \rangle \geq 0$ when $y = +1$ and $\langle \mathbf{w}, \mathbf{x} \rangle < 0$ when $y = -1$, where the variables are the coordinates $w_1, \ldots, w_d$ of $\mathbf{w}$. When we also consider a margin parameter $\gamma^* > 0$, then the constraints become $\langle \mathbf{w}, \mathbf{x} \rangle \geq \gamma^*$ and $\langle \mathbf{w}, \mathbf{x} \rangle < -\gamma^*$, respectively. Notice here that, for our purpose, we want (i) our halfspace $\mathbf{w}$ to be in $\mathbb{B}_1^d$, i.e., $|w_1| + \cdots + |w_d| \leq 1$, and (ii) each of our samples $\mathbf{x}$ to lie in $\mathbb{B}_\infty^d$, i.e., $|x_1|, \ldots, |x_d| \leq 1$.

Although the reductions in previous works vary in certain steps, they do share an overall common framework. With some simplification, they typically let e.g. $d = |U| \cdot |\Sigma_U|$, where each coordinate is associated with $U \times \Sigma_U$. In the completeness case, i.e., when some labeling $\phi^c$ covers all vertices in $V$, the intended solution $\mathbf{w}^c$ is defined by $w^c_{(u, \sigma_u)} = \mathbb{1}[\sigma_u = \phi(u)]/k$ for all $u \in U, \sigma_u \in \Sigma_U$. To ensure that this is essentially the best choice of halfspace, these reductions often appeal to several types of linear constraints. For concreteness, we state a simplified version of those from [ABSS97] below.

- For every $(u, \sigma_U) \in U \times \Sigma_U$, create the constraint $w_{(u, \sigma_u)} \leq 0$. (This corresponds to the labeled sample $(-\mathbf{e}_{(a, \sigma)}, +1)$.)

- For each $u \in U$, create the constraint $\sum_{\sigma \in \Sigma_U} w_{(u, \sigma)} \geq 1/k$.

- For every $v \in V$, $\sigma_v \in \Sigma_V$ and $u_1, u_2 \in N(v)$, add $\sum_{\sigma_{u_1} \in \pi^{-1}_{(u_1, v)}(\sigma_v)} w_{(u_1, \sigma_{u_1})} = \sum_{\sigma_{u_2} \in \pi^{-1}_{(u_2, v)}(\sigma_v)} w_{(u_2, \sigma_{u_2})}$. This equality "checks" the Label Cover constraints $\pi_{(u_1, v)}$ and $\pi_{(u_2, v)}$.

Clearly, in the completeness case $\mathbf{w}^c$ satisfies all constraints except the non-positivity constraints for the $k$ non-zero coordinates. (It was argued in [ABSS97] that any halfspace must violate many more constraints in the soundness case.) Observe that this reduction does not yield any margin: $\mathbf{w}^c$ does *not* classify any sample with a positive margin. Nonetheless, [DKM19] adapts this reduction to work with a small margin $\gamma^* > 0$ by adding/subtracting appropriate "slack" from each constraint. For example, the first type of constraint is changed to $w_{(u, \sigma_u)} \leq \gamma^*$. This gives the desired margin $\gamma^*$ in the completeness case. However, for the soundness analysis to work, it is crucial that $\gamma^* \leq O(1/k)$, as otherwise the constraints can be trivially satisfied[6] by $\mathbf{w} = \mathbf{0}$. As such, the above reduction does *not* work for us, since we would like a margin $\gamma^* = \Omega(1/\sqrt{k})$. In fact, this also holds for all known reductions, which are "local" in nature and possess similar characteristics. Roughly speaking, each linear constraint of these reductions involves only a constant number of terms that are intended to be set to $O(1/k)$, which means that we cannot hope to get a margin more than $O(1/k)$.

**Our Approach: Beyond Local Reductions.** With the preceding discussion in mind, our reduction has to be "non-local". To describe our main idea, we need an additional notion of "decomposability" of a Label Cover instance. Roughly speaking, an instance is *decomposable* if we can partition $V$ into different parts such that each $u \in U$ has exactly one induced edge to the vertices in each part.

**Definition 10.** *A Label Cover instance $\mathcal{L} = (U, V, E, \Sigma_U, \Sigma_V, \{\pi_e\}_{e \in E})$ is said to be decomposable if there exists a partition of $V$ into $V_1 \cup \cdots \cup V_t$ such that, for every $u \in U$ and $j \in [t]$, $|N(u) \cap V_j| = 1$. We use the notation $v^j(u)$ to the denote the unique element in $N(u) \cap V_j$.*

As explained above, "local" reductions use each labeled sample to only check a constant number of Label Cover constraints. In contrast, our reduction will check many constraints in each sample. Specifically, for each subset $V^j$, we will check all the Label Cover constraints involving $v \in V^j$ at once. To formalize this goal, we will require the following definition.

**Definition 11.** *Let $\mathcal{L} = (U, V = V_1 \cup \cdots \cup V_t, E, \Sigma_U, \Sigma_V, \{\pi_e\}_{e \in E})$ be a decomposable Label Cover instance. For any $j \in [t]$, let $\Pi^j \in \mathbb{R}^{(V \times \Sigma_V) \times (U \times \Sigma_U)}$ be defined as*

$$\Pi^j_{(v, \sigma_v), (u, \sigma_u)} = \begin{cases} 1 & \text{if } v = v^j(u) \text{ and } \pi_{(u,v)}(\sigma_u) = \sigma_v, \\ 0 & \text{otherwise.} \end{cases}$$

We set $d = |U| \cdot |\Sigma_U|$ and our intended solution $\mathbf{w}^c$ in the completeness case is the same as described in the previous reduction. For simplicity, suppose that, in the soundness case, we pick $\phi^s$ that does

*not* weakly cover any $v \in V$ and set $w^s_{(u,\sigma_u)} = \mathbb{1}[\sigma_u = \phi^s(u)]/k$. Our simplified task then becomes: *Design $\mathcal{D}$ such that $\mathrm{err}^{\mathcal{D}}_{\gamma}(\mathbf{w}^c) \ll \mathrm{err}^{\mathcal{D}}_{(1-\nu)\gamma}(\mathbf{w}^s)$, where $\gamma = \Omega(1/\sqrt{k})$, $\nu > 0$ is a constant.*

Our choice of $\mathcal{D}$ is based on two observations. The first is a structural difference between $\mathbf{w}^c(\Pi^j)^T$ and $\mathbf{w}^s(\Pi^j)^T$. Suppose that the constraint graph has right degree $\Delta$. Since $\phi^c$ covers all $v \in V$, $\Pi^j$ "projects" the non-zeros coordinates $w^c_{(u,\phi^c(u))}$ for all $u \in N(v)$ to the same coordinate $(v,\sigma_v)$, for some $\sigma_v \in \Sigma_V$, resulting in the value of $\Delta/k$ in this coordinate. On the other hand, since $\phi^s$ does not even weakly cover any right vertex, all the non-zero coordinates get maps by $\Pi^j$ to different coordinates, resulting in the vector $\mathbf{w}^s(\Pi^j)^T$ having $k$ non-zero coordinates, each having value $1/k$.

To summarize, we have: $\mathbf{w}^c(\Pi^j)^T$ has $k/\Delta$ non-zero coordinates, each of value $\Delta/k$. On the other hand, $\mathbf{w}^s(\Pi^j)^T$ has $k$ non-zero coordinates, each of value $1/k$.

Our second observation is the following: suppose that $\mathbf{u}$ is a vector with $T$ non-zero coordinates, each of value $1/T$. If we take a random $\pm 1$ vector $\mathbf{s}$, then $\langle \mathbf{u}, \mathbf{s} \rangle$ is simply $1/T$ times a sum of $T$ i.i.d. Rademacher random variables. Recall a well-known version of the central limit theorem (e.g., [Ber41, Ess42]): as $T \to \infty$, $1/\sqrt{T}$ times a sum of $T$ i.i.d. Rademacher r.v.s converges in distribution to the normal distribution. This implies that $\lim_{T \to \infty} \Pr[\langle \mathbf{u}, \mathbf{s} \rangle \geq 1/\sqrt{T}] = \Phi(1)$.

For simplicity, let us ignore the limit for the moment and assume that $\Pr[\langle \mathbf{u}, \mathbf{s} \rangle \geq 1/\sqrt{T}] = \Phi(1)$. We can now specify the desired distribution $\mathcal{D}$: Pick $\mathbf{s}$ uniformly at random from $\{\pm 1\}^{V \times \Sigma_V}$ and then let the sample be $\mathbf{s}\Pi^j$ with label $+1$. By the above two observations, $\mathbf{w}^c$ will be correctly classified with margin $\gamma^* = \sqrt{\Delta/k} = \Omega(1/\sqrt{k})$ with probability $\Phi(1)$. Furthermore, in the soundness case, $\mathbf{w}^s$ can only get the same error with margin (roughly) $\sqrt{1/k} = \gamma^*/\sqrt{\Delta}$. Intuitively, for $\Delta > 1$, this means that we get a gap of $\Omega(1/\sqrt{k})$ in the margins between the two cases, as desired. This concludes our informal proof overview.

**Further Details and The Full Reduction.** Having stated the rough main ideas above, we next state the full reduction. To facilitate this, we define the following additional notations:

**Definition 12.** *Let $\mathcal{L} = (U, V = V_1 \cup \cdots \cup V_t, E, \Sigma_U, \Sigma_V, \{\pi_e\}_{e \in E})$ be a decomposable Label Cover instance. For any $j \in [t]$, let $\hat{\Pi}^j \in \mathbb{R}^{(U \times \Sigma_V) \times (U \times \Sigma_U)}$ be such that*

$$\hat{\Pi}^j_{(u',\sigma_v),(u,\sigma_u)} = \begin{cases} 1 & \text{if } u' = u \text{ and } \pi_{(u,v^j(u))}(\sigma_u) = \sigma_v, \\ 0 & \text{otherwise}. \end{cases}$$

*Moreover, let $\tilde{\Pi}^j \in \mathbb{R}^{(V \times \Sigma_V) \times (U \times \Sigma_V)}$ be such that*

$$\tilde{\Pi}^j_{(v,\sigma'_v),(u,\sigma_v)} = \begin{cases} 1 & \text{if } v = v^j(u) \text{ and } \sigma'_v = \sigma_v \\ 0 & \text{otherwise}. \end{cases}$$

*Observe that $\Pi^j = \tilde{\Pi}^j \cdot \hat{\Pi}^j$ (where $\Pi^j$ is as in Definition 11).*

Our full reduction is present in Figure 1 below. The exact choice of parameters are deferred to the supplementary material. We note that the distribution described in the previous section corresponds to Step 4c in the reduction. The other steps of the reductions are included to handle certain technical details we had glossed over previously. In particular, the following are the two main additional technical issues we have to deal with here.

- *(Non-Uniformity of Weights)* In the intuitive argument above, we assume that, in the soundness case, we only consider $\mathbf{w}^s$ such that $\sum_{\sigma_u \in \Sigma_U} w^s_{(u,\sigma_u)} = 1/k$. However, this needs not be true in general, and we have to create new samples to (approximately) enforce such a condition. Specifically, for every subset $T \subseteq U$, we add a constraint that $\sum_{u \in T} \sum_{\sigma_u \in \Sigma_U} w_{(u,\sigma_u)} \geq |T|/k - \gamma^*$. This corresponds to Step 3 in Figure 1.
  Note that the term $-\gamma^*$ on the right hand side above is necessary to ensure that, in the completeness case, we still have a margin of $\gamma^*$. Unfortunately, this also leaves the possibility of, e.g., some vertex $u \in U$ has as much as $\gamma^*$ extra "mass". For technical reasons, it turns out that we have to make sure that these extra "masses" do not contribute to too much of $\|\mathbf{w}(\Pi^j)^T\|_2^2$. To do so, we add additional constraints on $\mathbf{w}(\hat{\Pi}^j)^T$ to bound its norm. Such a constraint is of the form: If we pick a subset $S$ of at most $\ell$ coordinates, then their sum must be at most $|S|/k + \gamma^*$ (and at least $-\gamma^*$). These corresponds to Steps 4a and 4b in Figure 1.

- *(Constant Coordinate)* Finally, similar to previous works, we cannot have "constants" in our linear constraints. Rather, we need to add a coordinate $\star$ with the intention that $\mathbf{w}_\star = 1/2$, and replace the constants in the previous step by $\mathbf{w}_\star$. Note here that we need two additional constraints (Steps 1 and 2 in Figure 1) to ensure that $\mathbf{w}_\star$ has to be roughly $1/2$.

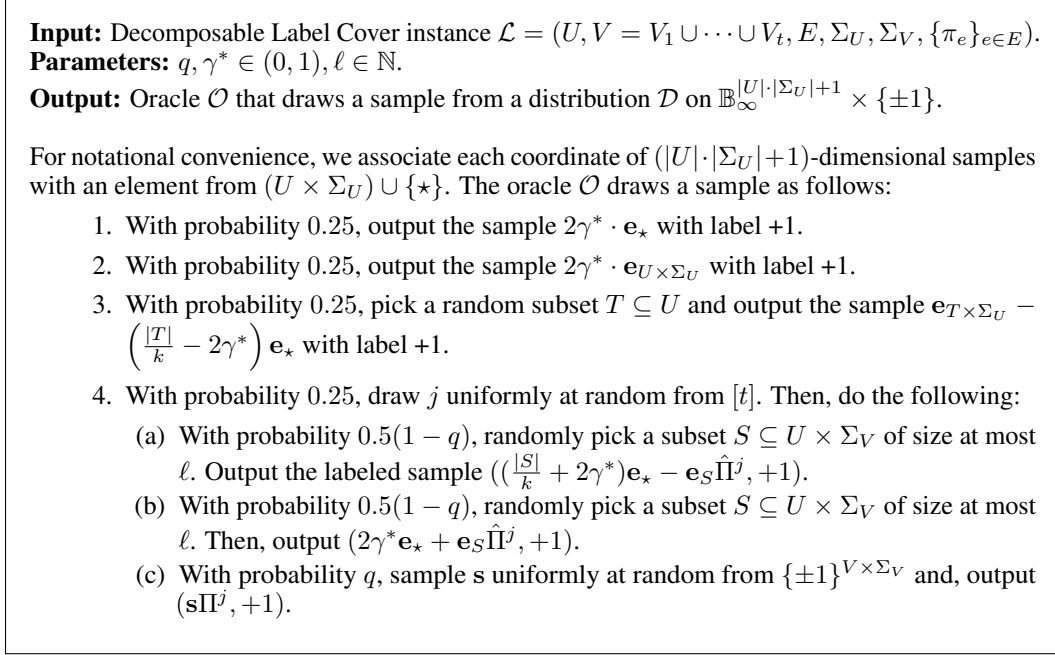

**Input:** Decomposable Label Cover instance $\mathcal{L} = (U, V = V_1 \cup \cdots \cup V_t, E, \Sigma_U, \Sigma_V, \{\pi_e\}_{e \in E})$.
**Parameters:** $q, \gamma^* \in (0, 1), \ell \in \mathbb{N}$.
**Output:** Oracle $\mathcal{O}$ that draws a sample from a distribution $\mathcal{D}$ on $\mathbb{B}_\infty^{|U| \cdot |\Sigma_U| + 1} \times \{\pm 1\}$.

For notational convenience, we associate each coordinate of $(|U| \cdot |\Sigma_U| + 1)$-dimensional samples with an element from $(U \times \Sigma_U) \cup \{\star\}$. The oracle $\mathcal{O}$ draws a sample as follows:

1. With probability 0.25, output the sample $2\gamma^* \cdot \mathbf{e}_\star$ with label +1.

2. With probability 0.25, output the sample $2\gamma^* \cdot \mathbf{e}_{U \times \Sigma_U}$ with label +1.

3. With probability 0.25, pick a random subset $T \subseteq U$ and output the sample $\mathbf{e}_{T \times \Sigma_U} - \left( \frac{|T|}{k} - 2\gamma^* \right) \mathbf{e}_\star$ with label +1.

4. With probability 0.25, draw $j$ uniformly at random from $[t]$. Then, do the following:

   (a) With probability $0.5(1 - q)$, randomly pick a subset $S \subseteq U \times \Sigma_V$ of size at most $\ell$. Output the labeled sample $((\frac{|S|}{k} + 2\gamma^*)\mathbf{e}_\star - \mathbf{e}_S \hat{\Pi}^j, +1)$.

   (b) With probability $0.5(1 - q)$, randomly pick a subset $S \subseteq U \times \Sigma_V$ of size at most $\ell$. Then, output $(2\gamma^* \mathbf{e}_\star + \mathbf{e}_S \hat{\Pi}^j, +1)$.

   (c) With probability $q$, sample $\mathbf{s}$ uniformly at random from $\{\pm 1\}^{V \times \Sigma_V}$ and, output $(\mathbf{s}\Pi^j, +1)$.

Figure 1: Hardness Reduction from Label Cover to $L_\infty$-margin Halfspace Learning. Here we use $\mathbf{e}_i$ to denote the $i$-th vector in the standard basis (i.e. the vector with value one in the $i$-th coordinate and zero in the remaining coordinates). Furthermore, we extend this notation and use $\mathbf{e}_S$, for a set $S$ of coordinates, to denote the indicator vector for $S$, i.e. $\mathbf{e}_S = \sum_{i \in S} \mathbf{e}_i$.

## 4    Conclusions and Open Problems

In this work, we studied the computational complexity of adversarially robust learning of halfspaces in the distribution-independent agnostic PAC model. We provided a simple proper learning algorithm for this problem and a nearly matching computational lower bound. While proper learners are typically preferable due to their interpretability, the obvious open question is whether significantly faster non-proper learners are possible. We leave this as an interesting open problem. Another direction for future work is to understand the effect of distributional assumptions on the complexity of the problem and to explore the learnability of simple neural networks in this context.

## Broader Impact

Our work aims to advance the algorithmic foundations of adversarially robust machine learning. This subfield focuses on protecting machine learning models (especially their predictions) against small perturbations of the input data. This broad goal is a pressing challenge in many real-world scenarios, where successful adversarial example attacks can have far-reaching implications given the adoption of machine learning in a wide variety of applications, from self-driving cars to banking.

Since the primary focus of our work is theoretical and addresses a simple concept class, we do not expect our results to have immediate societal impact. Nonetheless, we believe that our findings provide interesting insights on the algorithmic possibilities and fundamental computational limitations of adversarially robust learning. We hope that, in the future, these insights could be useful in the design of practically relevant adversarially robust classifiers in the presence of noisy data.

## Acknowledgments and Disclosure of Funding

Ilias Diakonikolas is supported by NSF Award CCF-1652862 (CAREER) and a Sloan Research Fellowship. Daniel M. Kane is supported by NSF Award CCF-1553288 (CAREER) and a Sloan Research Fellowship.

## Footnotes

[1]The function $\mathrm{sgn} : \mathbb{R} \to \{\pm 1\}$ is defined as $\mathrm{sgn}(u) = 1$ if $u \ge 0$ and $\mathrm{sgn}(u) = -1$ otherwise.

[2]Label Cover is sometimes referred to as *Projection Game* or *Two-Prover One-Round Game*.

[3]This is equivalent to $\pi_{(u_1,v)}(\phi(u_1)) = \pi_{(u_2,v)}(\phi(u_2))$ for all neighbors $u_1, u_2$ of $v$.

[4]For every $a \in U \cup V$, we use $N(a)$ to denote the set of neighbors of $a$ (with respect to the graph $(U, V, E)$).

[5]Some of these reductions are stated in terms of reductions from Set Cover or from constraint satisfaction problems (CSP). However, it is well-known that these can be formulated as Label Cover.

[6]Note that $\mathbf{w} = \mathbf{0}$ satisfies the constraints with margin $\gamma^* - 1/k$, which is $(1 - o(1))\gamma^*$ if $\gamma^* = \omega(1/k)$.

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
