[Supplementary Material]

# Supplementary Material

## A  Additional Background for Hardness Result

In this section, we provide additional preliminaries required for the proof of Theorem 3. Throughout the lower bound proof in the next section, we will sometimes view a vector $\mathbf{w} \in \mathbb{R}^d$ naturally as a column matrix $\mathbf{w} \in \mathbb{R}^{1 \times d}$; for example, we may write $\langle \mathbf{w}, \mathbf{x} \rangle = \mathbf{w}\mathbf{x}^T$. Furthermore, for any positive integer $m$, we use $[m]$ to denote $\{1, \ldots, m\}$. We also use $\mathbf{e}_i$ to denote the $i$-th vector in the standard basis (i.e., the vector with value one in the $i$-th coordinate and zero in the remaining coordinates). We extend this notation to a set $S$ of coordinates and use $\mathbf{e}_S$ to denote the indicator vector for $S$, i.e., $\mathbf{e}_S = \sum_{i \in S} \mathbf{e}_i$.

### A.1  Exponential Time Hypotheses

Recall that, in the 3-satisfiability (3SAT) problem, we are given a set of clauses, where each clause is an OR of at most three literals. The goal is to determine whether there exists an assignment that satisfies all clauses. The Exponential Time Hypothesis (ETH) [IP01, IPZ01] asserts that there is no sub-exponential time algorithm for 3SAT. ETH is of course a strengthening of the famous $P \neq NP$ assumption. In recent years, this assumption has become an essential part of modern complexity theory, as it allows one to prove tight running time lower bounds for many NP-hard and parameterized problems. See, e.g., [LMS11] for a survey on the topic.

For our lower bound, we use a strengthening of ETH, called Gap-ETH. Roughly speaking, Gap-ETH says that even finding an *approximate* solution to 3SAT is hard. This is stated more precisely below:

**Hypothesis 13** ((Randomized) Gap Exponential Time Hypothesis (Gap-ETH) [Din16, MR17])**.** *There exists a constant $\zeta > 0$ such that no randomized $2^{o(n)}$-time algorithm can, given a 3SAT instance on $n$ variables, distinguish between the following two cases correctly with probability $2/3$:*

- *(Completeness) There exists an assignment that satisfies all clauses.*

- *(Soundness) Every assignment violates at least $\zeta$ fraction of the clauses.*

Although proposed relatively recently, Gap-ETH is intimately related to a well-known open question whether linear size probabilistic checkable proofs exist for 3SAT; for more detail, please refer to the discussion in [Din16]. Gap-ETH has been used as a starting point for proving numerous tight running time lower bounds against approximation algorithms (e.g., [Din16, MR17, BGS17, AS18, JKR19]) and parameterized approximation algorithms (e.g., [CCK+17, DM18, BGKM18, CGK+19]). Indeed, we will use one such result as a starting point of our hardness reduction.

### A.2  Hardness of Label Cover

Recall the definition of Label Cover and its decomposibility from Definitions 8 and 10, respectively. We will use the following hardness of approximation result for Label Cover:

**Theorem 14** ([Man20])**.** *Assuming Gap-ETH, for any function $f$ and any constants $\Delta \in \mathbb{N} \setminus \{1\}, \mu \in (0, 1)$, there is no $f(k) \cdot n^{o(k)}$-time algorithm that can, given a decomposable Label Cover instance $\mathcal{L} = (U, V = V_1 \cup \cdots \cup V_t, E, \Sigma_U, \Sigma_V, \{\pi_e\}_{e \in E})$ whose right-degree is equal to $\Delta$, distinguish between*

- *(Completeness)* $\mathrm{val}(\mathcal{L}) = 1$,

- *(Soundness)* $\mathrm{wval}(\mathcal{L}) < \mu$,

*where $k := |U|$ and $n := |U| \cdot |\Sigma_U| + |V| \cdot |\Sigma_V|$.*

We remark here that the above theorem is not exactly the same as stated in [Man20]. We now briefly explain how to derive the version above from the one in [Man20]. Specifically, in [Man20], the decomposability of the instance $\mathcal{L}$ is not stated; rather, the instance there has the following property: $V$ is simply all subsets of size $\Delta$ of $U$, and, for any vertex $\{u_1, \ldots, u_\Delta\} \in V$, its neighbors are $u_1, \ldots, u_\Delta \in U$. Now, we can assume w.l.o.g. that $k$ is divisible by $\Delta$ by expanding each vertex

$u \in U$ to $\Delta$ new vertices $u^1, \ldots, u^\Delta$ and replicate each vertex in $\{u_1, \ldots, u_\Delta\} \in V$ to $\Delta^\Delta$ new vertices $\{u_1^{\xi(1)}, \ldots, u_\Delta^{\xi(\Delta)}\}$ for all $\xi : [\Delta] \to [\Delta]$. Once we have that $k$ is divisible by $\Delta$, Baranyai's theorem [Bar75] immediately implies the decomposability of the instance.

### A.3 Anti-Concentration

It is well-known that, if we take $m$ i.i.d. Rademacher random variables, their sum divided by $\sqrt{m}$ converges in distribution to the standard normal distribution (see, e.g., [Ber41, Ess42]). As a consequence, this immediately implies the following "anti-concentration" style result:

**Lemma 15.** *There exists $C \in (0, 1)$ and $m_0 > 0$ such that, for any $m \geq m_0$, we have*

$$\Pr_{X_1, \ldots, X_m} [X_1 + \cdots + X_m \geq C\sqrt{m}] \geq 0.4 \;,$$

*where $X_1, \ldots, X_m$ are i.i.d. Rademacher random variables.*

Note that the constant 0.4 above can be replaced by any constant strictly less than 0.5. We only use 0.4 here to avoid introducing additional variables.

## B    Proof of Tight Running Time Lower Bound for $L_\infty$ (Theorem 3)

Given the background from Section A, in this section we proceed to prove our computational lower bound (Theorem 3). As explained in the proof overview, the main ingredient of our hardness result is a reduction from Label Cover to the problem of $L_\infty$-$\gamma$-margin halfspace learning. The formal properties of our reduction are summarized below.

**Theorem 16** (Hardness Reduction)**.** *There exist absolute constants $\Delta, k_0 \in \mathbb{N} \setminus \{1\}$ and $\mu, \delta > 0$ such that the following holds. There is a polynomial time reduction that takes in a decomposable Label Cover instance $\mathcal{L} = (U, V = V_1 \cup \cdots \cup V_t, E, \Sigma_U, \Sigma_V, \{\pi_e\}_{e \in E})$ whose right-degree is equal to $\Delta$, and produces real numbers $\gamma^*, \epsilon^* > 0$ and an oracle $\mathcal{O}$ that can draw a sample from a distribution $\mathcal{D}$ on $\mathbb{B}_\infty^{|U| \cdot |\Sigma_U| + 1} \times \{\pm 1\}$ in polynomial time, such that when $|U| \geq k_0$ we have:*

- *(Completeness) If $\mathcal{L}$ is fully satisfiable (i.e., $\mathrm{val}(\mathcal{L}) = 1$), then $\mathrm{OPT}_{\gamma^*}^{\mathcal{D}} \leq \epsilon^*$.*

- *(Soundness) If $\mathrm{wval}(\mathcal{L}) < \mu$, then $\mathrm{OPT}_{(1-\delta)\gamma^*}^{\mathcal{D}} > 1.6\epsilon^*$.*

- *(Margin Bound) $\gamma^* \geq \Omega(1/\sqrt{k})$.*

- *(Error Bound) $\epsilon^* \geq n^{-O(\sqrt{k})}$.*

*Here $k := |U|$ and $n := |U| \cdot |\Sigma_U| + |V| \cdot |\Sigma_V|$ are defined similarly to Theorem 14.*

We remark that, similar to Theorem 3, the constant 1.6 in the soundness above can be changed to any constant strictly less than two. However, we choose to use an explicit constant here to avoid having a further variable.

The majority of this section will be spent proving Theorem 16. Specifically, in Section B.1, we describe the parameter setting for our reduction (including the margin bound and the error bound). Then, in Sections B.2 and B.3, we prove the completeness and the soundness respectively. Finally, in Section B.4, we briefly argue how it implies the claimed running time lower bound (Theorem 3).

### B.1    Parameter Selection for the Reduction

Recall that we already presented the reduction in Figure 1. The parameters of our reduction are chosen as follows:

- $C$ and $m_0$ are as in Lemma 15,
- $\Delta = \lceil 10^4/C^2 \rceil$,
- $\gamma^* = 0.5C\sqrt{\Delta/k}$,
- $k_0 = m_0\Delta$,

- $\delta = (0.1/\Delta)^4$,
- $\ell = \lceil \delta \sqrt{k} \rceil$,
- $q = 0.001/n^\ell$ (where $n$ is as defined is Theorem 14),
- $\epsilon^* = 0.6(0.25q)$,
- $\mu = \frac{0.01}{\Delta(\Delta-1)}$.

It is easy to see that the oracle can draw a sample in polynomial time. Furthermore, $\epsilon^* = 0.001/n^\ell \geq n^{-O(\sqrt{k})}$ and $\gamma^* \geq \Omega(1/\sqrt{k})$, as desired. Hence, we are only left to prove the completeness and the soundness of the reduction, which we will do next.

## B.2 Completeness

Suppose that the Label Cover instance $\mathcal{L}$ is satisfiable, i.e., that there exists a labeling $\phi^*$ that covers all right vertices. Let $\mathbf{w}^*$ be such that $w_\star^* = 1/2$ and

$$w_{(u,\sigma)}^* = \begin{cases} \frac{1}{2k} & \text{if } \sigma = \phi^*(u), \\ 0 & \text{otherwise} \end{cases}$$

for all $u \in U, \sigma \in \Sigma_U$. It is simple to check that the samples generated in Steps 1, 2, 3, 4a and 4b are all correctly labeled with margin $\gamma^*$.

Hence, we are left with computing the probability that the samples generated in Step 4c are violated. To do this, first notice that, for every $j \in [t], v \in V^j, \sigma_v \in \Sigma_V$, we have

$$(\mathbf{w}^*(\Pi^j)^T)_{(v,\sigma_v)} = \sum_{\substack{(u,\sigma_u) \in U \times \Sigma_U \\ v^j(u)=v, \pi_{(u,v)}(\sigma_u)=\sigma_v}} w_{(u,\sigma_u)}^*$$

$$(\text{From definition of } w^*) = \frac{1}{2k} \left| \{ u \in N(v) \mid \pi_{(u,v)}(\phi^*(u)) = \sigma_v \} \right|.$$

Now since every $v \in V^j$ is covered by $\phi^*$, there exists a unique $\sigma_v$ such that $\pi_{(u,v)}(\phi^*(u)) = \sigma_v$ for all $u \in N(v)$. As a result, $\mathbf{w}^*(\Pi^j)^T$ has $|V^j| = k/\Delta$ coordinates exactly equal to $\Delta \cdot \frac{1}{2k} = \frac{\Delta}{2k}$, and the remaining coordinates are equal to zero. Recall that, for the samples in Step 4c, $\mathbf{s}$ is a random $\{\pm 1\}$ vector. Thus, $\langle \mathbf{w}^*, \mathbf{s}\Pi^j \rangle = \langle \mathbf{w}^*(\Pi^j)^T, \mathbf{s} \rangle$ has the same distribution as $\frac{\Delta}{2k}$ times a sum of $k/\Delta$ i.i.d. Rademacher random variables. By Lemma 15, we can conclude that $\Pr_{\mathbf{s}}[\langle \mathbf{w}^*, \mathbf{s}\Pi^j \rangle \geq 0.5C\sqrt{\Delta/k}] \geq 0.4$. Since we set $\gamma^* = 0.5C\sqrt{\Delta/k}$, this implies that $\mathbf{w}^*$ correctly classifies (at least) 0.4 fraction of the samples from Step 4c. Hence, we have

$$\text{err}_\gamma^{\mathcal{D}}(\mathbf{w}^*) \leq 0.6 \cdot (0.25q) = \epsilon^*,$$

as desired.

## B.3 Soundness

We will prove the soundness contrapositively. For this purpose, suppose that there is a halfspace $\mathbf{w} \in \mathbb{B}_1^d$ such that $\text{err}_{(1-\delta)\gamma}^{\mathcal{D}}(\mathbf{w}) \leq 1.6\epsilon^* = 0.96(0.25q)$. We will show that there exists an assignments $\phi'$ with $\text{wval}(\phi') \geq \mu$.

### B.3.1 Some Simple Bounds

We start by proving a few observations/lemmas that will be useful in the subsequent steps.

First, observe that every distinct sample from Steps 1, 2, 3, 4a and 4b has probability mass (in $\mathcal{D}$) at least $\frac{0.125(1-q)}{n^\ell} > q > 1.6\epsilon^*$. Since we assume that $\text{err}_{(1-\delta)\gamma}^{\mathcal{D}}(\mathbf{w}) \leq 1.6\epsilon^*$, it must be the case that all these examples are correctly classified by $\mathbf{w}$ with margin at least $(1-\delta)\gamma^*$:

**Observation 17.** $\mathbf{w}$ *correctly classifies all samples in Steps 1, 2, 3, 4a and 4b with margin* $(1-\delta)\gamma^*$.

Throughout the remainder of this section, we will use the following notations:

**Definition 18.** *For every $u \in U$, let $M_u$ denote $\sum_{\sigma \in \Sigma_u} |w_{(u,\sigma)}|$. Then, let $U_{small}$ denote $\{u \in U \mid M_u \leq 1/k\}$ and $U_{large}$ denote $U \setminus U_{small}$.*

The next observation, which follows almost immediately from Observation 17, is that the value of the "constant coordinate" $w_\star$ is roughly $1/2$ (as we had in the completeness case) and that the sum of the absolute values of the negative coordinates is quite small.

**Observation 19.** *The following holds:*

1. *(Constant Coordinate Value) $w_\star \in [0.5(1 - \delta), 0.5(1 + \delta)]$.*

2. *(Negative Coordinate Value) $\sum_{\substack{j \in (U \times \Sigma_U) \cup \{\star\} \\ w_j < 0}} |w_j| \leq \delta$.*

*Proof.* 1. Since $\mathbf{w}$ correctly classifies the sample from Step 1 with margin $(1 - \delta)\gamma^*$, we have $2\gamma^* w_\star > (1 - \delta)\gamma^*$. This implies that $w_\star \geq 0.5(1 - \delta)$.

Let $a = \langle \mathbf{w}, \mathbf{e}_{U \times \Sigma_U} \rangle$. Similarly, from $\mathbf{w}$ correctly classifies the sample from Step 2 with margin $(1 - \delta)\gamma^*$, we have $a \geq 0.5(1 - \delta)$. Furthermore, observe that

$$a + w_\star \leq \|\mathbf{w}\|_1 \leq 1. \tag{3}$$

As a result, we have $w_\star \leq 0.5(1 + \delta)$ as desired.

2. Since $w_\star > 0$, we may rearrange the desired term as

$$\sum_{\substack{j \in (U \times \Sigma_U) \cup \{\star\} \\ w_j < 0}} |w_j| = \frac{1}{2} \left( \|\mathbf{w}\|_1 - a - w_\star \right)$$

$$\leq \frac{1}{2} \left( 1 - 0.5(1 - \delta) - 0.5(1 - \delta) \right)$$

$$< \delta,$$

where the first inequality follows from $a, w^* \geq 0.5(1 - \delta)$ that we had shown above. $\square$

Another bound we will use is that $U_{\text{large}}$ is quite small, and the sum of absolute values of the coordinates correspond to $U_{\text{large}}$ is also quite small.

**Observation 20** (Bounds on $U_{\text{large}}$)**.** *The following holds:*

1. *(Size Bound) $|U_{large}| \leq 2\delta k$.*

2. *(Mass Bound) $\sum_{u \in U_{large}} M_u \leq 2\delta$.*

*Proof.* To prove the desired bounds, first notice that, since $\mathbf{w}$ correctly classifies the sample in Step 3 with $T = U_{\text{small}}$ with margin $(1 - \delta)\gamma^*$, we must have

$$\langle \mathbf{w}, \mathbf{e}_{U_{\text{small}} \times \Sigma_U} \rangle \geq \left( \frac{|U_{\text{small}}|}{k} - 2\gamma^* \right) w_\star + (1 - \delta)\gamma^*.$$

Now, observe that the term on the left hand side is at most $\sum_{u \in U_{\text{small}}} M_u$ which, from $\|\mathbf{w}\|_1 \leq 1$, is in turn at most $1 - w_\star - \sum_{u \in U_{\text{large}}} M_u$. Combining these, we get

$$1 - w_\star - \sum_{u \in U_{\text{large}}} M_u \geq \left( \frac{|U_{\text{small}}|}{k} - 2\gamma^* \right) w_\star + (1 - \delta)\gamma^* = \left( 1 - \frac{|U_{\text{large}}|}{k} - 2\gamma^* \right) w_\star + (1 - \delta)\gamma^*$$

Recall from Observation 19 that $w_\star \geq 0.5(1 - \delta)$. Plugging this into the above, we have

$$\sum_{u \in U_{\text{large}}} M_u \leq 1 - \left( 2 - \frac{|U_{\text{large}}|}{k} - 2\gamma^* \right) \cdot 0.5(1 - \delta) - (1 - \delta)\gamma^*$$

$$= 1 - \left( 2 - \frac{|U_{\text{large}}|}{k} \right) \cdot 0.5(1 - \delta)$$

$$\leq \delta + \frac{0.5|U_{\text{large}}|}{k} . \tag{4}$$

1. Subtracting $\frac{0.5|U_{\text{large}}|}{k}$ from both sides, we have

$$\sum_{u \in U_{\text{large}}} \left(M_u - \frac{0.5}{k}\right) \leq \delta.$$

By definition, $M_u > 1/k$ for all $u \in U_{\text{large}}$. As a result, we have $|U_{\text{large}}| \leq 2\delta k$, as desired.

2. Plugging the bound on $|U_{\text{large}}|$ back into (4), we get the claimed bound on $\sum_{u \in U_{\text{large}}} M_u$. $\quad\square$

### B.3.2   Identifying a "Nice" Halfspace

We will now convert $\mathbf{w}$ into a "nicer" halfspace, i.e., one without negative and large coordinates. It will be much more convenient to deal with such a nice halfspace when we "decode" back a labeling later in this section.

The "nice" halfspace is quite simple: we just zero out all coordinates $w_{(u,\sigma)}$, where $u \in U_{\text{large}}$. More formally, let $\hat{\mathbf{w}} \in \mathbb{R}^{|U| \cdot |\Sigma_U|}$ be such that

$$\hat{w}_{(u,\sigma)} = \begin{cases} w_{(u,\sigma)} & u \in U_{\text{small}}, \\ 0 & u \in U_{\text{large}}, \end{cases}$$

for all $u \in U$ and $\sigma \in \Sigma_U$.

The main lemma needed in our analysis is that, for each $j \in [t]$, $\hat{\mathbf{w}}(\Pi^j)^T$ preserves most of the $L_2$ norm compared to the original $\mathbf{w}(\Pi^j)^T$.

**Lemma 21** (Nice Halfspace Preserves Most of $L_2$ Norm). *For every $j \in [t]$, we have*

$$\|\hat{\mathbf{w}}(\Pi^j)^T\|_2^2 \geq \frac{\|\mathbf{w}(\Pi^j)^T\|_2^2}{2} - \frac{\sqrt[4]{\delta}}{k}. \tag{5}$$

*Proof.* For convenience, let $\mathbf{v} = \mathbf{w} - \hat{\mathbf{w}}$ and $\mathbf{b} = \mathbf{v}(\hat{\Pi}^j)^T$. The majority of this proof is spent on bounding $\|\mathbf{b}\|_2^2$. To do this, let us define several new notations:

- Let $C_{>0} = |\{(u, \sigma) \in U \times \Sigma_V \mid b_{(u,\sigma)} > 0\}|$ and $C_{<0} = |\{(u, \sigma) \in U \times \Sigma_V \mid b_{(u,\sigma)} < 0\}|$.

- Let $\mathbf{b}^{\geq 0} \in \mathbb{R}^{U \times \Sigma_V}$ be defined by

$$b_{(u,\sigma)}^{\geq 0} = \max\{0, b_{(u,\sigma)}\}$$

for all $(u, \sigma) \in U \times \Sigma_V$. Furthermore, let $\mathbf{b}^{<0} = \mathbf{b} - \mathbf{b}^{\geq 0}$.

Observe that $\|\hat{\Pi}^j\|_1 \leq 1$, because each column has exactly a single entry equal to one and the remaining entries equal to zero. As a result, we have

$$\|\mathbf{b}\|_1 = \|\mathbf{v}(\hat{\Pi}^j)^T\|_1 \leq \|\mathbf{v}\|_1 = \sum_{u \in U_{large}} M_u \leq 2\delta, \tag{6}$$

where the last inequality follows from Observation 20.

Since $\mathbf{b} = \mathbf{b}^{\geq 0} + \mathbf{b}^{<0}$, we may bound $\|\mathbf{b}^{\geq 0}\|_2, \|\mathbf{b}^{<0}\|_2$ separately, starting with the former.

**Bounding $\|\mathbf{b}^{\geq 0}\|_2$.** Let us sort the coordinates of $\mathbf{b}^{\geq 0}$ from largest to smallest entries as $b_{(u^1,\sigma^1)}^{\geq 0}, \dots, b_{(u^{|U| \times |\Sigma_V|}, \sigma^{|U| \times |\Sigma_V|})}^{\geq 0}$ (tie broken arbitrarily). For every $j \leq \min\{C_{>0}, \ell\}$, consider the sample from Step 4a when $S = \{(u^1, \sigma^1), \dots, (u^j, \sigma^j)\}$. Since $\mathbf{w}$ correctly classifies this sample with margin $(1 - \delta)\gamma^*$, we have

$$(1 - \delta)\gamma^* \leq \left\langle \mathbf{w}, \left(\frac{j}{k} + 2\gamma^*\right)\mathbf{e}_\star - \mathbf{e}_S\hat{\Pi}^j\right\rangle$$

$$= \left(\frac{j}{k} + 2\gamma^*\right)w_\star - \mathbf{w}(\hat{\Pi}^j)^T(\mathbf{e}_S)^T$$

$$= \left(\frac{j}{k} + 2\gamma^*\right) w_\star - \left(\sum_{i \in [j]} (\mathbf{w}(\hat{\Pi}^j)^T)_{(u^i, \sigma^i)}\right)$$

$$\text{(Observation 19)} \leq \left(\frac{j}{k} + 2\gamma^*\right) \cdot 0.5(1+\delta) - \left(\sum_{i \in [j]} (\mathbf{w}(\hat{\Pi}^j)^T)_{(u^i, \sigma^i)}\right)$$

$$= \left(\frac{j}{k} + 2\gamma^*\right) \cdot 0.5(1+\delta) - \left(\sum_{i \in [j]} \left(b^{\geq 0}_{(u^i, \sigma^i)} + (\hat{\mathbf{w}}(\hat{\Pi}^j)^T)_{(u^i, \sigma^i)}\right)\right)$$

$$= \left(\frac{j}{k} + 2\gamma^*\right) \cdot 0.5(1+\delta) - \left(\sum_{i \in [j]} b^{\geq 0}_{(u^i, \sigma^i)}\right),$$

where the last equality follows from the fact that, for every $i \leq C_{>0}$, we must have $u^i \in U_{\text{large}}$ as otherwise $b^{\geq 0}_{(u^i, \sigma^i)}$ would have been equal to zero.

Rearranging the above inequality, we have

$$\left(\sum_{i \in [j]} b^{\geq 0}_{(u^i, \sigma^i)}\right) \leq \frac{0.5(1+\delta)j}{k} + 2\delta\gamma^* \leq \frac{j}{k} + 2\delta\gamma^*.$$

Recall from our assumption that $b^{\geq 0}_{(u^1, \sigma^1)} \geq \cdots \geq b^{\geq 0}_{(u^j, \sigma^j)}$. Plugging this into the above, we get

$$b^{\geq 0}_{(u^j, \sigma^j)} \leq \frac{1}{k} + \frac{2\delta\gamma^*}{j}. \tag{7}$$

Notice that while we have only derived the above inequality for $j \leq \min\{C_{>0}, \ell\}$, it also extends to all $j \leq \ell$ because $b^{\geq 0}_{(u^j, \sigma^j)} = 0$ for all $j > C_{>0}$.

We can use this to bound $\|\mathbf{b}^{\geq 0}\|_2^2$ as follows.

$$\|\mathbf{b}^{\geq 0}\|_2^2 = \sum_{j=1}^{|U| \cdot |\Sigma_V|} \left(b^{\geq 0}_{(u^j, \sigma^j)}\right)^2$$

$$= \sum_{j < \ell} \left(b^{\geq 0}_{(u^j, \sigma^j)}\right)^2 + \sum_{j \geq \ell} \left(b^{\geq 0}_{(u^j, \sigma^j)}\right)^2$$

$$\leq \sum_{j < \ell} \left(b^{\geq 0}_{(u^j, \sigma^j)}\right)^2 + b^{\geq 0}_{(u^\ell, \sigma^\ell)} \cdot \|\mathbf{b}^{\geq 0}\|_1$$

$$\overset{(7)}{\leq} \sum_{j < \ell} \left(\frac{1}{k} + \frac{2\delta\gamma^*}{j}\right)^2 + \left(\frac{1}{k} + \frac{2\delta\gamma^*}{\ell}\right) \cdot \|\mathbf{b}\|_1$$

$$\overset{(6)}{\leq} \sum_{j < \ell} 2\left(\frac{1}{k^2} + \frac{1}{j^2} \cdot 4\delta^2(\gamma^*)^2\right) + \left(\frac{1}{k} + \frac{2\delta\gamma^*}{\ell}\right) \cdot 2\delta$$

$$\leq \frac{2(\ell-1)}{k^2} + \frac{\pi^2}{6} \cdot 8\delta^2(\gamma^*)^2 + \frac{2\delta}{k} + \frac{4\delta^2\gamma^*}{\ell}$$

$$\text{(From our choice of } \ell \text{ and } \delta\gamma^* \leq 0.1\sqrt{\delta/k}) \leq \frac{2\delta}{k^{1.5}} + \frac{\delta}{k} + \frac{2\delta}{k} + \frac{\sqrt{\delta}}{k}$$

$$\leq \frac{2\sqrt{\delta}}{k}.$$

**Bounding $\|\mathbf{b}^{<0}\|_2$.** This is very similar (and in fact slightly simpler) to how we bound $\|\mathbf{b}^{\geq 0}\|_2$ above; we repeat the argument here for completeness. Let us first sort the coordinates of $\mathbf{b}^{<0}$ from smallest to largest entries as $b^{<0}_{(u^{-1}, \sigma^{-1})}, \ldots, b^{<0}_{(u^{-|U| \times |\Sigma_V|}, \sigma^{-|U| \times |\Sigma_V|})}$ (tie broken arbitrarily). For

every $j \le \min\{C_{<0}, \ell\}$, consider the sample from Step 4b when $S = \{(u^{-1}, \sigma^{-1}), \ldots, (u^{-j}, \sigma^{-j})\}$. Since $\mathbf{w}$ correctly classifies this sample with margin $(1 - \delta)\gamma^*$, we have

$$(1 - \delta)\gamma^* \le \left\langle \mathbf{w}, 2\gamma^* \mathbf{e}_\star + \mathbf{e}_S \hat{\Pi}^j \right\rangle$$

$$= 2\gamma^* \cdot w_\star + \mathbf{b}(\mathbf{e}_S)^T$$

$$\text{(Observation 19)} \le 2\gamma^* \cdot 0.5(1 + \delta) - \left( \sum_{i \in [j]} |b^{<0}_{(u^{-i}, \sigma^{-i})}| \right).$$

Rearranging the above inequality, we have

$$\left( \sum_{i \in [j]} |b^{<0}_{(u^{-i}, \sigma^{-i})}| \right) \le 2\delta\gamma^*.$$

Recall from our assumption that $|b^{<0}_{(u^{-1}, \sigma^{-1})}| \ge \cdots \ge |b^{<0}_{(u^{-j}, \sigma^{-j})}|$. Plugging this into the above, we get

$$|b^{<0}_{(u^{-j}, \sigma^{-j})}| \le \frac{2\delta\gamma^*}{j}. \tag{8}$$

Similar to the previous case, although we have derived the above inequality for $j \le \min\{C_{<0}, \ell\}$, it also holds for all $j \le \ell$ simply because $b^{<0}_{(u^{-j}, \sigma^{-j})} = 0$ for all $j > C_{<0}$.

We can use this to bound $\|\mathbf{b}^{<0}\|_2^2$ as follows.

$$\|\mathbf{b}^{<0}\|_2^2 = \sum_{j=1}^{|U| \cdot |\Sigma_V|} \left( b^{<0}_{(u^{-j}, \sigma^{-j})} \right)^2$$

$$= \sum_{j < \ell} \left( b^{<0}_{(u^{-j}, \sigma^{-j})} \right)^2 + \sum_{j \ge \ell} \left( b^{<0}_{(u^{-j}, \sigma^{-j})} \right)^2$$

$$\le \sum_{j < \ell} \left( b^{<0}_{(u^{-j}, \sigma^{-j})} \right)^2 + |b^{<0}_{(u^{-\ell}, \sigma^{-\ell})}| \cdot \|\mathbf{b}^{<0}\|_1$$

$$\overset{(8)}{\le} \sum_{j < \ell} \left( \frac{2\delta\gamma^*}{j} \right)^2 + \left( \frac{2\delta\gamma^*}{\ell} \right) \cdot \|\mathbf{b}\|_1$$

$$\overset{(6)}{\le} \frac{\pi^2}{6} \cdot 4\delta^2(\gamma^*)^2 + \left( \frac{2\delta\gamma^*}{\ell} \right) \cdot 2\delta$$

$$\text{(From our choice of } \ell \text{ and } \delta\gamma^* \le 0.1\sqrt{\delta/k}) \le \frac{\delta}{k} + \frac{\sqrt{\delta}}{k}$$

$$\le \frac{2\sqrt{\delta}}{k}.$$

Using our bounds on $\|\mathbf{b}^{\ge 0}\|_2^2, \|\mathbf{b}^{<0}\|_2^2$, we can easily bound $\|\mathbf{b}\|_2^2$ by

$$\|\mathbf{b}\|_2^2 = \|\mathbf{b}^{\ge 0}\|_2^2 + \|\mathbf{b}^{<0}\|_2^2 \le \frac{4\sqrt{\delta}}{k}. \tag{9}$$

Next observe that $\|\tilde{\Pi}^j\|_1 = 1$, because each column has exactly a single entry equal to one and the remaining entries equal to zero. Furthermore, $\|\tilde{\Pi}^j\|_\infty = \Delta$ because each row has exactly $\Delta$ entries equal to one[7]. As a result, by Holder's inequality, we have $\|\tilde{\Pi}^j\|_2 \le \sqrt{\|\tilde{\Pi}^j\|_1 \|\tilde{\Pi}^j\|_\infty} = \sqrt{\Delta}$. From this and from (9), we arrive at

$$\frac{4\sqrt{\delta} \cdot \Delta}{k} \ge \|\mathbf{b}(\tilde{\Pi}^j)^T\|_2^2 = \|\mathbf{v}(\Pi^j)^T\|_2^2, \tag{10}$$

where the latter follows from our definition of $\mathbf{b}$.

Thus, we have

$$\|\mathbf{w}(\Pi^j)^T\|_2^2 = \|\hat{\mathbf{w}}(\Pi^j)^T + \mathbf{v}(\Pi^j)^T\|_2^2 \leq 2\|\hat{\mathbf{w}}(\Pi^j)^T\|_2^2 + 2\|\mathbf{v}(\Pi^j)^T\|_2^2 \overset{(10)}{\leq} 2\|\hat{\mathbf{w}}(\Pi^j)^T\|_2^2 + \frac{8\sqrt{\delta}\Delta}{k}.$$

Finally, recall from our choice of parameter that $\sqrt{\delta}\Delta \leq 0.1\sqrt[4]{\delta}$. This, together with the above inequality, implies the desired bound. $\qquad\square$

### B.3.3 Decoding Label Cover Assignment

We now arrive at the last part of the proof, where we show that there exists an assignment that weakly covers at least $\mu = \frac{0.01}{\Delta(\Delta-1)}$ fraction of vertices in $V$, which completes our soundness proof.

**Lemma 22.** *There exists an assignment $\phi'$ of $\mathcal{L}$ such that* $\mathrm{wval}(\phi') \geq \mu$.

*Proof.* We define a (random) assignment $\phi$ for $\mathcal{L}$ as follows:

- For each $u \in U_{\mathrm{small}}$, let $\phi(u)$ be a random element from $\Sigma_U$ where $\sigma_u \in \Sigma_U$ is selected with probability $\frac{|\hat{w}_{(u,\sigma_u)}|}{\sum_{\sigma \in \Sigma_U} |\hat{w}_{(u,\sigma)}|}$.

- For each $u \in U_{\mathrm{large}}$, let $\phi(u)$ be an arbitrary element in $\Sigma_U$.

We will now argue that $\mathbb{E}_\phi[\mathrm{wval}(\phi)] \geq \mu$. Since we assume that $\mathrm{OPT}^{\mathcal{D}}_{(1-\delta)\gamma^*}(\mathbf{w}) \leq 0.96(0.25q)$, we have

$$\begin{aligned}
0.96(0.25q) &\geq \mathrm{OPT}^{\mathcal{D}}_{(1-\delta)\gamma^*}(\mathbf{w}) \\
&\geq (0.25q) \Pr_{j \in [t], \mathbf{s} \in \{\pm 1\}^{V \times \Sigma_V}} \left[\left\langle \mathbf{w}, \mathbf{s}\Pi^j \right\rangle < (1-\delta)\gamma^* \right],
\end{aligned}$$

where the second inequality is due to the error from the samples from Step 4c.

Let $J \subseteq [t]$ contain all $j \in [t]$ such that $\Pr_{\mathbf{s} \in \{\pm 1\}^{V \times \Sigma_V}} \left[\left\langle \mathbf{w}, \mathbf{s}\Pi^j \right\rangle < (1-\delta)\gamma^* \right] < 0.99$. The above inequality implies that

$$\Pr_{j \in [t]} [j \in J] > 0.01. \tag{11}$$

Now, let us fix $j \in J$. By definition of $J$, we have

$$\begin{aligned}
0.01 &\leq \Pr_{\mathbf{s} \in \{\pm 1\}^{V \times \Sigma_V}} \left[\left\langle \mathbf{w}, \mathbf{s}\Pi^j \right\rangle \geq (1-\delta)\gamma^* \right] \\
&\leq \Pr_{\mathbf{s} \in \{\pm 1\}^{V \times \Sigma_V}} \left[|\left\langle \mathbf{w}, \mathbf{s}\Pi^j \right\rangle| \geq (1-\delta)\gamma^* \right] \\
&= \Pr_{\mathbf{s} \in \{\pm 1\}^{V \times \Sigma_V}} \left[|\left\langle \mathbf{w}(\Pi^j)^T, \mathbf{s} \right\rangle|^2 \geq ((1-\delta)\gamma^*)^2 \right] \\
\text{(Markov's inequality)} \quad &\leq \frac{\mathbb{E}_{\mathbf{s} \in \{\pm 1\}^{V \times \Sigma_V}}[|\left\langle \mathbf{w}(\Pi^j)^T, \mathbf{s} \right\rangle|^2]}{((1-\delta)\gamma^*)^2} \\
&= \frac{\|\mathbf{w}(\Pi^j)^T\|_2^2}{((1-\delta)\gamma^*)^2}.
\end{aligned}$$

As a result, we must have $\|\mathbf{w}(\Pi^j)^T\|_2^2 \geq 0.01((1-\delta)\gamma^*)^2$. We now apply Lemma 21, which yields

$$\|\hat{\mathbf{w}}(\Pi^j)^T\|_2^2 \geq 0.005((1-\delta)\gamma^*)^2 - \frac{\sqrt[4]{\delta}}{k} \geq \frac{2}{k}. \tag{12}$$

Using the definition of $\Pi^j$, we may now rewrite $\|\hat{\mathbf{w}}(\Pi^j)^T\|_2^2$ as follows.

$$\|\hat{\mathbf{w}}(\Pi^j)^T\|_2^2$$

$$= \sum_{(v,\sigma_v)\in V\times\Sigma_V} ((\hat{\mathbf{w}}(\Pi^j)^T)_{(v,\sigma_v)})^2$$

$$= \sum_{(v,\sigma_v)\in V_j\times\Sigma_V} ((\hat{\mathbf{w}}(\Pi^j)^T)_{(v,\sigma_v)})^2$$

$$= \sum_{(v,\sigma_v)\in V_j\times\Sigma_V} \left( \sum_{u\in N(v),\sigma_u\in\pi^{-1}_{(u,v)}(\sigma_v)} \hat{w}_{(u,\sigma_u)} \right)^2$$

$$= \sum_{(v,\sigma_v)\in V_j\times\Sigma_V} \sum_{u\in N(v)} \left( \sum_{\sigma_u\in\pi^{-1}_{(u,v)}(\sigma_v)} \hat{w}_{(u,\sigma_u)} \right)^2$$

$$+ \sum_{\substack{(v,\sigma_v)\in V_j\times\Sigma_V \\ u,u'\in N(v) \\ u\neq u'}} \left( \sum_{\sigma_u\in\pi^{-1}_{(u,v)}(\sigma_v)} \hat{w}_{(u,\sigma_u)} \right) \left( \sum_{\sigma_{u'}\in\pi^{-1}_{(u',v)}(\sigma_v)} \hat{w}_{(u',\sigma_{u'})} \right)$$

$$= \sum_{(v,\sigma_v)\in V_j\times\Sigma_V} \sum_{u\in N(v)\cap U_{\text{small}}} \left( \sum_{\sigma_u\in\pi^{-1}_{(u,v)}(\sigma_v)} \hat{w}_{(u,\sigma_u)} \right)^2$$

$$+ \sum_{\substack{(v,\sigma_v)\in V_j\times\Sigma_V \\ u,u'\in N(v)\cap U_{\text{small}} \\ u\neq u'}} \left( \sum_{\sigma_u\in\pi^{-1}_{(u,v)}(\sigma_v)} \hat{w}_{(u,\sigma_u)} \right) \left( \sum_{\sigma_{u'}\in\pi^{-1}_{(u',v)}(\sigma_v)} \hat{w}_{(u',\sigma_{u'})} \right), \quad (13)$$

where the last equality follows from the fact that $\hat{w}_{(u,\sigma_u)} = 0$ for all $u\notin U_{\text{small}}$.

We will now bound the two terms in (13) separately. For the first term, we have

$$\sum_{(v,\sigma_v)\in V_j\times\Sigma_V} \sum_{u\in N(v)\cap U_{\text{small}}} \left( \sum_{\sigma_u\in\pi^{-1}_{(u,v)}(\sigma_v)} \hat{w}_{(u,\sigma_u)} \right)^2 \leq \sum_{(v,\sigma_v)\in V_j\times\Sigma_V} \sum_{u\in N(v)\cap U_{\text{small}}} \left( \sum_{\sigma_u\in\pi^{-1}_{(u,v)}(\sigma_v)} |\hat{w}_{(u,\sigma_u)}| \right)^2$$

$$= \sum_{u\in U_{\text{small}}} \left( \sum_{\sigma_v\in\Sigma_V} \left( \sum_{\sigma_u\in\pi^{-1}_{(u,v^j(u))}(\sigma_v)} |\hat{w}_{(u,\sigma_u)}| \right)^2 \right)$$

$$\leq \sum_{u\in U_{\text{small}}} \left( \sum_{\sigma_u\in\Sigma_U} |\hat{w}_{(u,\sigma_u)}| \right)^2$$

$$= \sum_{u\in U_{\text{small}}} M_u^2$$

$$\leq \frac{1}{k}, \quad (14)$$

where the last inequality follows from $M_u \leq 1/k$ for all $u \in U_{\text{small}}$ (by definition) and from $\sum_{u\in U_{\text{small}}} M_u \leq \|\mathbf{w}\|_1 \leq 1$.

We now move on to bound the second term of (13). To do so, let us observe that, for every $u\in U_{\text{small}}, v\in N(u)$ and $\sigma_v\in\Sigma_V$, we have

$$\Pr_{\phi}[\pi_{(u,v)}(\phi(u))=\sigma_v] = \sum_{\sigma_u\in\pi^{-1}_{(u,v)}(\sigma_v)} \frac{|\hat{w}_{(u,\sigma_u)}|}{M_u}$$

$$\geq k \sum_{\sigma_u\in\pi^{-1}_{(u,v)}(\sigma_v)} |\hat{w}_{(u,\sigma_u)}|.$$

As a result, we have

$$
\sum_{(v,\sigma_v)\in V_j\times\Sigma_V}\sum_{\substack{u,u'\in N(v)\cap U_{\text{small}}\\u\neq u'}}\left(\sum_{\sigma_u\in\pi_{(u,v)}^{-1}(\sigma_v)}\hat{w}_{(u,\sigma_u)}\right)\left(\sum_{\sigma_{u'}\in\pi_{(u',v)}^{-1}(\sigma_v)}\hat{w}_{(u',\sigma_{u'})}\right)
$$

$$
\leq\sum_{(v,\sigma_v)\in V_j\times\Sigma_V}\sum_{\substack{u,u'\in N(v)\cap U_{\text{small}}\\u\neq u'}}\frac{\Pr_\phi[\pi_{(u,v)}(\phi(u))=\sigma_v]}{k}\cdot\frac{\Pr_\phi[\pi_{(u',v)}(\phi(u'))=\sigma_v]}{k}
$$

$$
=\frac{1}{k^2}\sum_{(v,\sigma_v)\in V_j\times\Sigma_V}\sum_{\substack{u,u'\in N(v)\cap U_{\text{small}}\\u\neq u'}}\Pr_\phi[\pi_{(u,v)}(\phi(u))=\pi_{(u',v)}(\phi(u'))=\sigma_v]
$$

$$
=\frac{1}{k^2}\sum_{v\in V_j}\sum_{\substack{u,u'\in N(v)\cap U_{\text{small}}\\u\neq u'}}\sum_{\sigma_v\in\Sigma_V}\Pr_\phi[\pi_{(u,v)}(\phi(u))=\pi_{(u',v)}(\phi(u'))=\sigma_v]
$$

$$
=\frac{1}{k^2}\sum_{v\in V_j}\sum_{\substack{u,u'\in N(v)\cap U_{\text{small}}\\u\neq u'}}\Pr_\phi[\pi_{(u,v)}(\phi(u))=\pi_{(u',v)}(\phi(u'))]
$$

$$
\leq\frac{1}{k^2}\sum_{v\in V_j}\sum_{\substack{u,u'\in N(v)\cap U_{\text{small}}\\u\neq u'}}\Pr_\phi[\phi\text{ weakly covers }v]
$$

$$
\leq\frac{\Delta(\Delta-1)}{k^2}\sum_{v\in V_j}\Pr_\phi[\phi\text{ weakly covers }v]\,,\tag{15}
$$

where the last inequality follows from the fact that each $v\in V$ has degree $\Delta$.

Combining (12), (13), (14) and (15), we have

$$
\sum_{v\in V_j}\Pr_\phi[\phi\text{ weakly covers }v]\geq\frac{k}{\Delta(\Delta-1)}\,.
$$

By summing over all $j\in J$ and using the bound from (11), we have

$$
0.01t\cdot\frac{k}{\Delta(\Delta-1)}\leq\sum_{j\in J}\sum_{v\in V_j}\Pr_\phi[\phi\text{ weakly covers }v]
$$

$$
\leq\sum_{v\in V}\Pr_\phi[\phi\text{ weakly covers }v]
$$

$$
=|V|\cdot\mathbb{E}_\phi[\text{wval}(\phi)]
$$

$$
\leq kt\cdot\mathbb{E}_\phi[\text{wval}(\phi)]\,.
$$

Equivalently, this means that $\mathbb{E}_\phi[\text{wval}(\phi)]\geq\mu$, which implies that there exists an assignment $\phi'$ of $\mathcal{L}$ such that $\text{wval}(\phi')\geq\mu$, as desired. □

## B.4 Putting Things Together

Now that we have proved Theorem 16, we briefly argue that it implies the desired running time lower bound (Theorem 3).

*Proof of Theorem 3.* Let $\Delta,k_0,\mu,\delta$ be as in Theorem 16. Suppose for the sake of contradiction that there exists a proper $\delta$-robust 1.5-agnostic learner for $L_\infty$-$\gamma$-margin halfspace $\mathcal{A}$ that runs in time $f(1/\gamma)\cdot d^{o(1/\gamma^2)}\text{poly}(1/\epsilon)$. We will use this to construct an algorithm $\mathcal{B}$ for Label Cover.

Given a Label Cover instance $\mathcal{L}$ as an input, the algorithm $\mathcal{B}$ works as follows:

- Run the reduction from Theorem 16 on input $\mathcal{L}$ to get $\epsilon^*,\gamma^*,\mathcal{O}$.

- Run $\mathcal{A}$ on $\mathcal{O}$ with parameters $\gamma=\gamma^*,\epsilon=0.05\epsilon^*,\tau=0.9$ to get a halfspace $\mathbf{w}$.

- Draw $10^6/\epsilon^2$ additional samples from $\mathcal{O}$. Let $\tilde{\mathcal{D}}$ be the empirical distribution.

- If $\mathrm{err}^{\tilde{\mathcal{D}}}_{(1-\delta)\gamma}(\mathbf{w}) \leq 1.58\epsilon^*$, return YES. Otherwise, return NO.

The first step of $\mathcal{B}$ runs in $\mathrm{poly}(n)$ time. The second step runs in time $f(1/\gamma)d^{o(1/\gamma^2)}\,\mathrm{poly}(1/\epsilon) = f(O(\sqrt{k})) \cdot n^{o(k)} \cdot \mathrm{poly}(n^{O(\sqrt{k})}) = f(O(\sqrt{k})) \cdot n^{o(k)}$. The last two steps run in time $\mathrm{poly}(n, 1/\epsilon)$; recall from Theorem 16 that $\epsilon^* = n^{O(\sqrt{k})}$, meaning that these two steps run in time $n^{O(\sqrt{k})}$. Hence, the entire algorithm $\mathcal{B}$ runs in $g(k) \cdot n^{o(k)}$ time for some function $g$.

We will next argue the following correctness guarantee of the algorithm: If $\mathrm{val}(\mathcal{L}) = 1$, then the algorithm answers YES with probability 0.8 and, if $\mathrm{wval}(\mathcal{L}) < \mu$, then the algorithm returns NO with probability 0.8. Before we do so, observe that this, together with Theorem 14, means that Gap-ETH is violated, which would complete our proof.

Note that we may assume that $k \geq k_0$, as otherwise the Label Cover instance can already be solved in polynomial time. Now consider the case $\mathrm{val}(\mathcal{L}) = 1$. Theorem 16 ensures that $\mathrm{OPT}^{\mathcal{D}}_{\gamma^*} \leq \epsilon^*$. As a result, $\mathcal{A}$ returns $\mathbf{w}$ that satisfies the following with probability 0.9: $\mathrm{err}^{\mathcal{D}}_{(1-\delta)\gamma^*}(\mathbf{w}) \leq 1.5\epsilon^* + 0.05\epsilon^* = 1.55\epsilon^*$. Furthermore, it is simple to check that $\Pr[|\mathrm{err}^{\mathcal{D}}_{(1-\delta)\gamma^*}(\mathbf{w}) - \mathrm{err}^{\tilde{\mathcal{D}}}_{(1-\delta)\gamma^*}(\mathbf{w})| > 0.02\epsilon^*] \leq 0.1$. Hence, with probability 0.8, we must have $\mathrm{err}^{\tilde{\mathcal{D}}}_{(1-\delta)\gamma^*} \leq 1.57\epsilon^*$ and the algorithm returns YES.

On the other hand, suppose that $\mathrm{wval}(\mathcal{L}) < \mu$. The soundness of Theorem 16 ensures that $\mathrm{err}^{\mathcal{D}}_{(1-\delta)\gamma^*}(\mathbf{w}) > 1.6\epsilon^*$. Similar to before, since $\Pr[|\mathrm{err}^{\mathcal{D}}_{(1-\delta)\gamma^*}(\mathbf{w}) - \mathrm{err}^{\tilde{\mathcal{D}}}_{(1-\delta)\gamma^*}(\mathbf{w})| > 0.02\epsilon^*] \leq 0.1$, we have $\mathrm{err}^{\tilde{\mathcal{D}}}_{(1-\delta)\gamma^*}(\mathbf{w}) > 1.58\epsilon^*$ with probability at least 0.9. Thus, in this case, the algorithm returns NO with probability 0.9 as desired. $\qquad\square$

## C  Tight Running Time Lower Bound for $L_p$ when $2 \leq p < \infty$

In this section, we briefly sketch a nearly tight running time lower bound when $p \geq 2$ is a (finite) constant, which follows almost directly from our previous work [DKM19]. In fact, [DKM19] already provided such a hardness in the case of $p = 2$, which we summarize below.

**Theorem 23** ([DKM19]). *For any constant $\alpha > 1$, assuming Gap-ETH, there is no proper 1-robust $\alpha$-agnostic learner for $L_2$-$\gamma$-margin halfspace that runs in time $\mathrm{poly}(d/\epsilon) \cdot 2^{(1/\gamma)^{2-o(1)}}$.*

*Moreover, this holds even when $\gamma, \epsilon = 1/d^{O(1)}$.*

Note that the hardness above is quite strong in the two aspects. First, it holds even against 1-robust learner, i.e. when the learner only requires to output a halfspace with small classification error (with margin 0). Secondly, the above hardness holds for any constant factor $\alpha > 1$.

We can extend this hardness to any constant $p \geq 2$, which matches with our algorithm from Theorem 2 upto a $O(\gamma^{o(1)})$ factor in the exponent.

**Theorem 24.** *For any finite constant $p \geq 2$ and any constant $\alpha > 1$, assuming Gap-ETH, there is no proper 1-robust $\alpha$-agnostic learner for $L_p$-$\gamma$-margin halfspace that runs in time $\mathrm{poly}(d/\epsilon) \cdot 2^{(1/\gamma)^{2-o(1)}}$.*

In the proof sketch below, since we are dealing with multiple $L_p$ norms at once, we will write $\mathrm{err}^{\mathcal{D}}_{p,\gamma}$ to emphasize that this refers to $\mathrm{err}^{\mathcal{D}}_{\gamma}$ with respect to the $L_p$ norm.

*Proof Sketch.* Suppose for the sake of contradiction that, for some finite constant $p \geq 2$ and constants $\alpha > 1, \zeta' > 0$, there is a proper 1-robust $\alpha$-agnostic learner $\mathcal{A}$ for $L_p$-$\gamma'$-margin halfspace that runs in time $\mathrm{poly}(d/\epsilon') \cdot 2^{O\left((1/\gamma')^{2-\zeta'}\right)}$. We will use this to devise a proper 1-robust $\alpha$-agnostic learner $\mathcal{B}$ for $L_2$-$\gamma$-margin halfspace that runs in time $\mathrm{poly}(d/\epsilon) \cdot 2^{O\left((1/\gamma)^{2-\zeta}\right)}$ for some constant $\zeta > 0$ when $\gamma, \epsilon = 1/d^{O(1)}$. Together with Theorem 23, we arrive at the desired lower bound.

On input samples $(\mathbf{x}_1, y_1), \ldots, (\mathbf{x}_m, y_m)$, $\mathcal{B}$ works as follows:

- Sample a rotation matrix $\Pi \in \mathbb{R}^{d \times d}$ uniformly at random.

- Run $\mathcal{A}$ on samples $\left( \frac{\Pi \mathbf{x}_1}{\|\Pi \mathbf{x}_1\|_p}, y_1 \right), \ldots, \left( \frac{\Pi \mathbf{x}_m}{\|\Pi \mathbf{x}_m\|_p}, y_m \right)$ to get a halfspace $\mathbf{w}'$ where $\gamma' = \frac{\gamma}{C \cdot \log^2(\alpha/\epsilon)}$ and $\epsilon' = \epsilon/2$.

- Output $\frac{\Pi^{-1} \mathbf{w}'}{\|\Pi^{-1} \mathbf{w}'\|_q}$.

The claimed running time of $\mathcal{B}$ follows immediately from that of $\mathcal{A}$. We will now argue the correctness of $\mathcal{B}$. Let $\mathcal{D}'$ denote the distribution of $\left( \frac{\Pi \mathbf{x}}{\|\Pi \mathbf{x}\|_p}, y \right)$ where $(\mathbf{x}, y) \sim \mathcal{D}$. Our main technical claim is that when the constant $C$ (for $\gamma'$) is sufficiently large, the following holds with probability 0.9:

$$\mathrm{OPT}^{\mathcal{D}'}_{p,\gamma'} \leq \mathrm{OPT}^{\mathcal{D}}_{2,\gamma} + \frac{\epsilon}{2\alpha}. \tag{16}$$

Before we sketch the proof of (16), let us briefly argue the correctness assuming (16). Observe that $\mathrm{err}^{\mathcal{D}}_{2,0} \left( \frac{\Pi^{-1} \mathbf{w}'}{\|\Pi^{-1} \mathbf{w}'\|_q} \right) = \mathrm{err}^{\mathcal{D}'}_{p,0}(\mathbf{w}')$. As a result, we have

$$\mathrm{err}^{\mathcal{D}}_{2,0} \left( \frac{\Pi^{-1} \mathbf{w}'}{\|\Pi^{-1} \mathbf{w}'\|_q} \right) = \mathrm{err}^{\mathcal{D}'}_{p,0}(\mathbf{w}') \leq \alpha \cdot \mathrm{OPT}^{\mathcal{D}'}_{p,\gamma'} + \epsilon' \overset{(16)}{\leq} \alpha \cdot \mathrm{OPT}^{\mathcal{D}}_{2,\gamma} + \epsilon,$$

where the first inequality follows from the guarantee of $\mathcal{A}$. Thus, $\mathcal{B}$ is the desired $\alpha$-agnostic learner for $L_2$-$\gamma$-margin halfspace.

Finally, we turn our attention to proving (16). Let $\mathbf{w}^*$ be the optimal halfspace for $\mathcal{D}$, i.e. that $\mathrm{err}^{\mathcal{D}}_{2,\gamma}(\mathbf{w}^*) = \mathrm{OPT}^{\mathcal{D}}_{2,\gamma}$. It is simple to see that, when $C_1$ is a sufficiently large constant, we have $\|\Pi \mathbf{w}^*\|_q \leq C_1 \cdot d^{1/q - 1/2} \cdot \log(\alpha/\epsilon)$ with probability $1 - 0.01\epsilon/\alpha$. Similarly, for any $(\mathbf{x}, y) \in \mathrm{supp}(\mathcal{D})$, when $C_2$ is a sufficiently large constant we have $\|\Pi \mathbf{x}\|_p \leq C_2 \cdot d^{1/p - 1/2} \cdot \log(\alpha/\epsilon)$ with probability $1 - 0.01\epsilon/\alpha$. Let $C = 2C_1 C_2$. When both of these occur and $y \langle \mathbf{w}^*, \mathbf{x} \rangle \geq \gamma$, we have $y \left\langle \frac{\Pi \mathbf{w}^*}{\|\Pi \mathbf{w}^*\|_q}, \frac{\Pi \mathbf{x}}{\|\Pi \mathbf{x}\|_p} \right\rangle > \gamma'$. In other words, for each sample $(\mathbf{x}, y) \in \mathcal{D}$ correctly classified by $\mathbf{w}^*$ with margin $\gamma$ w.r.t. $L_2$ norm, $\left( \frac{\Pi \mathbf{x}}{\|\Pi \mathbf{x}\|_p}, y \right)$ is correctly classified by $\frac{\Pi \mathbf{w}^*}{\|\Pi \mathbf{w}^*\|_q}$ with margin $\gamma'$ w.r.t. $L_p$ norm with probability $1 - 0.02\epsilon/\alpha$. Markov's inequality then implies that, with probability 0.9, we have

$$\mathrm{OPT}^{\mathcal{D}'}_{p,\gamma'} \leq \mathrm{err}^{\mathcal{D}'}_{p,\gamma'} \left( \frac{\Pi \mathbf{w}^*}{\|\Pi \mathbf{w}^*\|_q} \right) \leq \mathrm{err}^{\mathcal{D}}_{2,\gamma}(\mathbf{w}^*) + \frac{\epsilon}{2\alpha} = \mathrm{OPT}^{\mathcal{D}}_{2,\gamma} + \frac{\epsilon}{2\alpha},$$

which concludes our proof sketch. $\qquad\square$

## D  Proof of Fact 1

*Proof of Fact 1.* For convenience, let $\mathbf{w}' = \frac{\mathbf{w}}{\|\mathbf{w}\|_q}$. Consider any $(\mathbf{x}, y) \in \mathbb{R}^d \times \{\pm 1\}$. We claim that $\mathrm{sgn}(\langle \mathbf{w}', \mathbf{x} \rangle - \gamma) \neq y$ iff $\exists \mathbf{z} \in \mathcal{U}_{p,\gamma}(\mathbf{x}), h_{\mathbf{w}}(\mathbf{z}) \neq y$. Below we only show this statement when $y = -1$. The case $y = 1$ follows analogously.

Suppose $y = -1$. Let us first prove the forward direction: if $\mathrm{sgn}(\langle \mathbf{w}', \mathbf{x} \rangle - \gamma) \neq y = -1$, we have $\langle \mathbf{w}', \mathbf{x} \rangle \geq \gamma$. Let $\mathbf{t} \in \mathbb{R}^d$ be such that $t_i = \gamma \cdot \mathrm{sgn}(w_i') \cdot |w_i'|^{q-1}$. It is simple to verify that $\|\mathbf{t}\|_p = \gamma$ and that $\langle \mathbf{w}', \mathbf{t} \rangle = \gamma$. Consider $\mathbf{z} = \mathbf{x} - \mathbf{t} \in \mathcal{U}_{p,\gamma}(\mathbf{x})$. We have

$$\langle \mathbf{w}', \mathbf{z} \rangle = \langle \mathbf{w}', \mathbf{x} \rangle - \langle \mathbf{w}', \mathbf{t} \rangle \geq 0.$$

Thus, we have $h_{\mathbf{w}}(\mathbf{z}) = h_{\mathbf{w}'}(\mathbf{z}) = 1 \neq y$ as desired.

We will next prove the converse by contrapositive. Suppose that $\mathrm{sgn}(\langle \mathbf{w}', \mathbf{x} \rangle - \gamma) = y = -1$. Then, we have $\langle \mathbf{w}', \mathbf{x} \rangle < -\gamma$ and, for any $\mathbf{z} \in \mathcal{U}_{p,\gamma}(\mathbf{x})$, we can derive

$$\langle \mathbf{w}', \mathbf{z} \rangle \leq \langle \mathbf{w}', \mathbf{x} \rangle + |\langle \mathbf{w}', \mathbf{z} - \mathbf{x} \rangle|$$
$$\text{(Holder's Inequality)} < -\gamma + \|\mathbf{w}'\|_q \|\mathbf{z} - \mathbf{x}\|_p$$
$$< 0,$$

where the last inequality follows from $\|\mathbf{w}'\|_q = 1$ and $\|\mathbf{z} - \mathbf{x}\|_p \leq \gamma$. Hence, $h_{\mathbf{w}}(\mathbf{z}) = h_{\mathbf{w}'}(\mathbf{z}) = -1 = y$ as desired.

To summarize, so far we have shown that $\mathrm{sgn}(\langle \mathbf{w}', \mathbf{x} \rangle - \gamma) \neq y$ iff $\exists \mathbf{z} \in \mathcal{U}_{p,\gamma}(\mathbf{x}), h_{\mathbf{w}}(\mathbf{z}) \neq y$. As a result, we have

$$\mathcal{R}_{\mathcal{U}_{p,\gamma}}(h_{\mathbf{w}}, \mathcal{D}) = \Pr_{(\mathbf{x},y)\sim\mathcal{D}} [\exists \mathbf{z} \in \mathcal{U}_{p,\gamma}(\mathbf{x}), h_{\mathbf{w}}(\mathbf{z}) \neq y]$$

$$= \Pr_{(\mathbf{x},y)\sim\mathcal{D}} [\mathrm{sgn}(\langle \mathbf{w}', \mathbf{x} \rangle - \gamma) \neq y]$$

$$= \mathrm{err}_\gamma^{\mathcal{D}}(\mathbf{w}') \,. \qquad \qquad \square$$

## E  On the Necessity of Bicriterion Approximation

In this section, we briefly argue that, when there is no margin gap (i.e., for $\nu = 0$), the learning problem we consider is computationally hard. In particular, we show the following hardness that, when $\nu = 0$ and[8] $\gamma = 0.5$, there is no $\mathrm{poly}(d/\epsilon)$-time learning algorithm for any constant approximation ratio $\alpha > 1$. Note that this result holds under the assumption $NP \nsubseteq RP$. If we further assume ETH, we can get a stronger lower bound of $2^{(d/\epsilon)^c}$ for some constant $c > 0$. This is in contrast to our main algorithmic result (Theorem 2) that, when $\nu, \gamma > 0$ and $\alpha > 1$ are constants, runs in polynomial (in $d/\epsilon$) time.

**Proposition 25.** *For any constant $\alpha > 1$, assuming $NP \nsubseteq RP$, there is no proper 0-robust $\alpha$-agnostic learner for $L_\infty$-0.5-margin halfspaces in time $\mathrm{poly}(d/\epsilon)$.*

Similar to before (see, e.g., Section 3), the above result immediately follows from Lemma 26 below. We will henceforth focus on the proof of this lemma.

**Lemma 26.** *For any constant $\alpha > 1$, assuming $P \neq NP$, no $\mathrm{poly}(d/\epsilon)$-time algorithm can, given $\epsilon > 0$ and a multiset $S \subseteq \mathbb{B}_\infty^d \times \{\pm 1\}$ of labeled samples, distinguish between:*

- *(Completeness) $\mathrm{OPT}_{0.5}^S \leq \epsilon$.*

- *(Soundness) $\mathrm{OPT}_{0.5}^S > \alpha \cdot \epsilon$.*

To prove Lemma 26, we will use the following hardness for (no-margin) proper agnostic learning of halfspaces. Observe here that in the Completeness case, there is an extra promise that every coordinate of $w$ is non-negative; this follows from the construction of [ABSS97].

**Theorem 27** ([ABSS97]). *For any constant $\alpha > 1$, assuming $P \neq NP$, no $\mathrm{poly}(\tilde{d}/\tilde{\epsilon})$-time algorithm can, given $\tilde{\epsilon} > 0$ and a multiset $\tilde{S} \subseteq \mathbb{B}_\infty^{\tilde{d}} \times \{\pm 1\}$ of labeled samples, distinguish between:*

- *(Completeness) There exists $\tilde{\mathbf{w}} \in \mathbb{B}_1^{\tilde{d}}$ where $\tilde{w}_i \geq 0$ for all $i \in [d]$ such that $\mathrm{err}_0^{\tilde{S}}(\tilde{\mathbf{w}}) \leq \tilde{\epsilon}$.*

- *(Soundness) $\mathrm{OPT}_0^{\tilde{S}} > \alpha \cdot \tilde{\epsilon}$.*

*Proof of Lemma 26.* Given a multiset $\tilde{S} \subseteq \mathbb{B}_\infty^{\tilde{d}} \times \{\pm 1\}$ from Theorem 27. Let $m = |\tilde{S}|$. We create a new multiset of samples $S \subseteq \mathbb{B}_\infty^d \times \{\pm 1\}$ as follows:

- Let $d = \tilde{d} + 1$.

- For every $(\mathbf{x}, y) \in \tilde{S}$, add[9] $(\mathbf{x} \circ y, y)$ to $S$.

- Add $\lceil \alpha m + 1 \rceil$ copies of $((1, \dots, 1, 0), +1)$ to $S$.

Finally, let $\epsilon = \frac{\tilde{\epsilon} \cdot m}{m + \lceil \alpha m + 1 \rceil}$. It is obvious that the reduction runs in polynomial time. We will now argue its completeness and soundness.

**Completeness.** Suppose that there is $\tilde{\mathbf{w}} \in \mathbb{B}_1^{\tilde{d}}$ whose coordinates are non-negative such that $\mathrm{err}_0^{\tilde{S}}(\tilde{\mathbf{w}}) \leq \tilde{\epsilon}$. Consider $\mathbf{w} = (0.5\tilde{\mathbf{w}}/\|\tilde{\mathbf{w}}\|_1) \circ 0.5$. Since each coordinate of $\tilde{\mathbf{w}}$ is non-negative, the new halfspace $\mathbf{w}$ correctly classifies the last sample with margin 0.5. Furthermore, it is also simple to verify that $(\mathbf{x}, y) \in \tilde{S}$ is correctly classified by $\tilde{\mathbf{w}}$ (with margin 0) iff $(\mathbf{x} \circ y, y)$ is correctly classified by $\mathbf{w}$ with margin 0.5. As a result, we have $\mathrm{err}_{0.5}^{S}(\mathbf{w}) = \frac{m}{m+\lceil\alpha m+1\rceil} \cdot \mathrm{err}_0^{\tilde{S}}(\tilde{\mathbf{w}}) \leq \epsilon$, as desired.

**Soundness.** Suppose that $\mathrm{OPT}_0^{\tilde{S}} > \alpha \cdot \tilde{\epsilon}$. Consider any $\mathbf{w} \in \mathbb{B}_1^{\tilde{d}}$. Let us consider two cases, based on the value of $w_{d+1}$.

- $w_{d+1} > 1/2$. In this case, $\langle \mathbf{w}, (1, \ldots, 1, 0) \rangle < 0.5$. In other words, $\mathbf{w}$ does *not* correctly classify the last sample with margin 0.5. As a result, we immediately have $\mathrm{err}_{0.5}^{S}(\mathbf{w}) \geq \frac{\lceil\alpha m+1\rceil}{m+\lceil\alpha m+1\rceil} > \alpha \cdot \epsilon$ as desired.

- $w_{d+1} \leq 1/2$. In this case, notice that $\mathrm{sgn}(\langle \mathbf{w}, \mathbf{x} \circ y \rangle - 0.5y) = y$ implies that $\mathrm{sgn}(\langle (w_1, \ldots, w_d), \mathbf{x} \rangle) = y$. Thus, we have $\mathrm{err}_{0.5}^{S}(\mathbf{w}) \geq \frac{m}{m+\lceil\alpha m+1\rceil} \cdot \mathrm{err}_0^{\tilde{S}}((w_1, \ldots, w_d)) \geq \frac{m}{m+\lceil\alpha m+1\rceil} \cdot (\alpha \cdot \tilde{\epsilon}) = \alpha \cdot \epsilon$.

Hence, in both cases, we have $\mathrm{OPT}_{0.5}^{S} > \alpha \cdot \epsilon$, which concludes our proof. $\qquad\square$

## F  Additional Open Questions

In addition to the broader open questions posed in Section 4, we list several concrete open questions below, regarding our lower bound (Theorem 3).

- As alluded to in Section 4, our proof can only rule out a margin gap $(\gamma, (1 - \nu)\gamma)$ when $\nu > 0$ is a small constant. An intriguing direction here is to extend our hardness to include a larger $\nu$, or conversely give a better algorithm for larger $\nu$. We remark that even the case of margin gap $(\gamma, 0)$ (i.e., $\nu = 1$) remains open for the $L_\infty$-margin setting. In this case, the learner only seeks a small misclassification error (without any margin). Note that [DKM19] gave hardness results that hold even when $\nu = 1$ in the setting of $L_2$-margin.

- Our technical approach can rule out approximation ratio $\alpha$ of at most 2. The reason is that, the labeled samples (Step 4c in our reduction) that test the Label Cover constraints are still violated with probability at least 0.5 by the intended solution. As a result, any "reasonable" solution will achieve an approximation ratio of 2. In contrast, [DKM19] can rule out any constant $\alpha$. Can our hardness be strengthened to also handle larger values of $\alpha$?

- Finally, it may be interesting to attempt to prove our hardness result under a weaker assumption, specifically ETH. Note that this is open for both our $L_\infty$-margin setting and the $L_2$-margin setting in [DKM19][10]. This question is closely related to the general research direction of basing parameterized inapproximability results under ETH instead of Gap-ETH. There are some parameterized hardness of approximation results known under ETH (e.g., [Mar13, CL19, KLM19, Lin19, BBE$^+$19]), but a large number of questions remain open, including basing Theorem 14 on ETH instead of Gap-ETH, which would have given our hardness of $L_\infty$-margin learning under ETH. However, it might be possible to give a different proof for hardness of $L_\infty$-margin learning assuming ETH directly, without going through such a result as Theorem 14.

## Footnotes

[7]For every row $(v, \sigma_v)$, these 1-entries are the entries $(u, \sigma_v)$ for all $u \in N(v)$.

[8]We remark that 0.5 is unimportant here and the reduction works for any $\gamma \leq 0.5$.

[9]Note that we use $\mathbf{x} \circ y$ to denote the vector resulting from concatenating $\mathbf{x}$ and $y$.

[10]In [DKM19], the hardness result is stated under ETH but it is not asymptotically tight (as there is a $\gamma^{o(1)}$ factor in the exponent); their reduction only gives asymptotically tight hardness under Gap-ETH.