[Reviews · NeurIPS 2020]

Review 1

Summary and Contributions: This paper studies the problem of agnostic learning a halfspace that is adversarially robust to L_p perturbations (in the agnostic PAC model). For halfspaces, this is equivalent to learning agnostic proper learning a halfspace that minimizes the \gamma-margin error, where the margin in measured in some L_p norm. The main results are: 1. An algorithm that runs in time exp(p / \gamma^2)*poly(d) that incurs a small (1+\eps) approximation in both the error, and in the margin. This also gets good guarantees for L_\infty norm (as usual, you can think of p ~ log d here for L_\infty). 2. They show that this dependence of the form d^{O(1/\gamma^2)} for L_\infty norm is unavoidable assuming the Gap-ETH assumption. The algorithm is using a very elegant reduction from online mistake-bounded learning. The algorithmic result was shown in the special case of L_2 [DKM19] using other techniques.

Strengths: The topic of the paper is timely and well-motivated; we do not understand basic questions involving (computational) efficiently learning adversarial robust classifiers. The results in this paper represent a solid contribution. Perhaps more interesting are the techniques used in obtaining these results. The reduction from online mistake-bounded learning is clever and very elegant. It immediately implies the correct bounds for different norms, while bounds were only known for L_2 using more complicated arguments. Moreover, the hardness result assuming ETH involves fairly sophisticated PCP machinery that might use for other problems (even in fine-grained complexity). Overall, I enjoyed reading the paper, and learning from it.

Weaknesses: The techniques in this paper are somewhat specific to halfspaces which are very simple hypotheses (adversarial robustness does not correspond to a standard L_p margin for other concept classes). But that being said, it's important to understand these basic hypothesis classes first.

Correctness: I did not read enough of the appendix to verify the details hardness proof. But whatever I did read looked correct, and I have no reasons to doubt the correctness.

Clarity: The paper is very well written.

Relation to Prior Work: Yes

Reproducibility: Yes

Additional Feedback: A minor comment for the authors: I wonder if you could use these arguments to show guarantees based on the perceptron algorithm itself. The perceptron analysis shows that there is a combination of O(1/gamma^2) of the examples that gives a good classifier. I wonder if one can also show a similar statement with the margin error (with some loss in margin), and guess the examples in the combination. Perhaps these are already implied based on the mistake-bound learning reduction(?). AFTER AUTHOR RESPONSE Thanks for the clear response. I like the paper; I'm sticking with my score of 8.


Review 2

Summary and Contributions: This paper studies the problem of adversarially robust (proper) learning of halfspaces in the agnostic case, with respect to L_p perturbations. This paper furthers our understanding of what we can / cannot do in adversarially robust learning, for the simple class of halfspaces. The contributions of the paper are as follows: 1. A computationally "efficient" learning algorithm that handles the case of L_p perturbations for all p >= 2, including p = infinity. 2. A hardness result showing that in p=infinity case, the running time of the algorithm is nearly optimal (at least in some parameters) assuming the "Gap Exponential Time Hypothesis". This shows the limits of the best results we could hope for without making any distributional assumptions. RESPONSE TO AUTHORS: Thank you for the response. My rating for the paper is unchanged, and I continue to feel the paper should be accepted.

Strengths: The paper introduces new techniques to establish their results, which might be useful beyond the context of this paper: - To give the learning algorithm, the paper introduces a simple reduction from online learning to agnostic PAC learning. I think this reduction could be useful beyond the setting studied in this paper. - The hardness result proved here reduces from the standard problem of 'Label Cover'. However, there are many subtle details to make the proof go through. In particular, the reduction constructed here has to be "non-local", which goes beyond prior works which primarily used "local" reductions.

Weaknesses: I find the hardness result a bit limited in that it only show that the dependence in terms of d and \gamma is tight in the case of L_{\infty} perturbations. The result is stated for a "small constant" \nu > 0. Perhaps it might help to say how small \nu needs to be for the result to hold. For example, would \nu=0.1 work?

Correctness: The proofs seem correct to me at a high level; there are many details in the hardness result, which I did not fully verify.

Clarity: The paper is well written.

Relation to Prior Work: The relation to prior work is adequately discussed.

Reproducibility: Yes

Additional Feedback: Minor comments: - Line 13: The citation [Colter-Madry18] should be [Kolter-Madry '18]. - Line 153 (Lemma 6): It might be clearer to spell out what the "polynomial time" is. Isn't it poly(|S|, M, d)?


Review 3

Summary and Contributions: The authors study adversarially robust semi-agnostic learning of halfspaces. Their work is built on the observation that learning halfspaces in a manner robust to adversarial L_p perturbations is essentially equivalent to learning halfspaces with an L_p margin. Hence, the authors are able to make blackbox use of existing algorithms. The main algorithmic difficulty seems to be that since the learning should also be agnostic, we want to avoid selecting a halfspace that has been influenced by labeled examples incompatible with the margin guarantee for the optimal halfspace. To circumvent this difficulty, the authors use an existing *online* algorithm for learning halfspaces with margin, combined with many random attempts to iteratively select a set of "good" examples from the sample set. The authors also give essentially matching lower bounds for the case of L_infty perturbations, under the Gap-ETH hypothesis. This is proved via a reduction from Label Cover instances to the learning problem.

Strengths: The lower bound seems fairly novel. Compared to previous reductions from Label Cover, the authors are able to get substantially better margin bounds by exploiting global "decomposability" structure in the hard Label Cover instances. This provides a good illustration of the limits for agnostic adversially robust learning, even in such a simple setting as halfspaces.

Weaknesses: The algorithmic upper bound seems to be a fairly straightforward application of existing halfspace learning algorithms; the main contribution here is realizing that the existing online learners are enough. Furthermore, there are a number of other papers giving adversarially robust learning guarantees for halfspaces, including (non-agnostic) learning for L_p perturbations with random classification noise, and semi-agnostic learning for L_2 perturbations.

Correctness: Yes, though I think the summary presented for the lower bound could be clearer. Specifically, it would be nice to note that the hard instances for Theorem 9 indeed have the decomposability structure.

Clarity: Yes, the exposition is reasonably clear, the notation is not overwhelming.

Relation to Prior Work: Yes, in both Sections 1 and 3 there is very nice context concerning existing works.

Reproducibility: Yes

Additional Feedback: I thank the authors for their response. I agree that the simplicity of the upper bound proof is an advantage.


Review 4

Summary and Contributions: This paper extends and complements recent work by Diakonikolas et al. on essentially the same problem: supposing that there exists a halfspace that only fails to get margin gamma on an OPT fraction of points, find a halfspace that only fails to get margin (1+nu)gamma on a (1+delta)OPT+epsilon fraction. As noted in the introduction, if the examples lie in the l_p unit ball and the margin is calculated over weights in the l_q unit ball for q dual to p, this corresponds to agnostic learning of a classifier that is robust to l_p perturbations. The main (positive) result of Diakonikolas et al. is restricted to p=q=2, and features an exponential dependence on 1/delta; here the result is generalized to all p, and the dependence on delta is improved to exponential in log(1/delta) (i.e., it is moved to the base of the exponent). (Prior recent works had treated the case of proper robust learning.) Both works also prove some negative results. Diakonikolas et al. gave a hardness result for the l_2 norm under ETH, showing that for some constant nu and any constant delta, the exponential dependence on 1/gamma^2 of all of these algorithms is essentially necessary. This work gives a lower bound for the l_infinity norm, showing that the dimension^O(1/gamma^2) term in the running time obtained by their algorithm for the l_infinity norm problem is similarly necessary.

Strengths: The problem of perturbation robustness is of significant interest at the moment, and l_2 perturbations are not the most natural kinds of perturbations to consider in that context -- indeed, l_infinity is the most commonly considered, and this is the main case treated by this work. Moreover, although the realizable case had been treated by previous work, some kind of noise tolerance is essential in practice, and in some ways agnostic noise tolerance is the ideal guarantee, but also the hardest to achieve. Especially in the context of the complementary running time lower bound, this seems like a good result in this direction. Also, the algorithm here is structured as a generic reduction to a variant of mistake bounded learning with a margin gap, for which an algorithm existed previously in the literature. This is a nice conceptual connection between the models. Finally, the construction of the lower bound has some nice techniques, and the body of the submission does a good job of summarizing the technical innovation here.

Weaknesses: I don't want to overstate the weaknesses, they are relatively minor. Nevertheless, given the amount of work on closely related problems, this work is in some ways a little incremental. Also, one might argue that a stochastic noise model is often of more relevance in practice than the highly pessimistic agnostic noise model, where much stronger guarantees are generally possible for stochastic noise. (I feel that the agnostic model is still worthy of study and more relevant for some scenarios, though.) Also there was no empirical evaluation, but I feel that the theoretical results are sufficiently interesting to merit publication. Response to authors: Sure, Massart noise is an excellent example of a realistic noise model. (It is not quite as pessimistic as agnostic noise, of course...)

Correctness: Yes, I feel convinced by the arguments in the paper.

Clarity: Yes. Although the lower bound is pretty technical, there is a very nice overview of the arguments.

Relation to Prior Work: Yes, there is a fair discussion of the various works in this area putting this work in context.

Reproducibility: Yes

Additional Feedback:

[Author Response · NeurIPS 2020]

We thank the reviewers for their careful reading of the paper. Please find the answers to the questions below.

**Review #1**

*Q: The techniques in this paper are somewhat specific to halfspaces which are very simple hypotheses (adversarial robustness does not correspond to a standard $L_p$ margin for other concept classes). But that being said, it's important to understand these basic hypothesis classes first.*

A: As the reviewer states, halfspaces form one of the most basic and fundamental hypothesis class in machine learning, and we believe that obtaining a complete understanding of adversarial robustness for halfspaces is an important contribution. Furthermore, the hardness results for such a basic concept class might also be a good indication that the problem is hard for other, more complicated, classes too.

*Q: I wonder if you could use these arguments to show guarantees based on the perceptron algorithm itself. The perceptron analysis shows that there is a combination of $O(1/\gamma^2)$ of the examples that gives a good classifier. I wonder if one can also show a similar statement with the margin error, and guess the examples in the combination.*

A: The reviewer's suggestion is correct; a slight modification of the perceptron algorithm, where an update is performed whenever a sample is not correctly classified with margin $(1 - \nu)\gamma$, is known to be an $L_2$ online learner with margin gap $(\gamma, (1 - \nu)\gamma)$ and mistake bound $O\left(\frac{1}{\nu^2\gamma^2}\right)$. When plugging this so-called "margin perceptron" algorithm to our reduction (Proposition 5), this recovers our theorem in the case $p = 2$. Indeed, the more general algorithm of [Gen01a] that we invoke (Theorem 7) can be viewed as a slight modification of margin perceptron in the case $p = 2$.

**Review #2**

*Q: I find the hardness result a bit limited in that it only show that the dependence in terms of $d, \gamma$ is tight in the case $L_\infty$.*

A: As we briefly mentioned in the paper, the (essentially) tight running time lower bound of the form $2^{\gamma^{1-o(1)}}$ for $L_p$ perturbations for any constant $p \geq 2$ follows already from [DKM19]. Specifically, [DKM19] proved such a result for $p = 2$. To get a similar lower bound for other constants $p \geq 2$, we simply take a random rotation of every sample $\mathbf{x}$ (of the hard instance for $L_2$ perturbation) and rescale so that it has unit $L_p$ norm while keeping the label the same as before. (The optimal halfspace is also rotated and scaled so that it has unit $L_q$ norm.) It is not hard to see that this preserves the margin for most of the samples up to a constant factor. We will add more detail about this in the revised version.

*Q: The result is stated for a "small constant" $\nu > 0$. Perhaps it might help to say how small $\nu$ needs to be for the result to hold. For example, would $\nu = 0.1$ work?*

A: We agree that obtaining a concrete value of $\nu$ is interesting; in fact, this is included in our "Additional Open Questions" in the supplementary material. For our current proof, $\nu$ is selected to be very tiny ($\approx 10^{-16}$) for simplicity of presentation. Per rough estimates, we can have $\nu \approx 10^{-4}$ but the bounds in the proof become more delicate.

**Review #3**

*Q: The algorithmic upper bound seems to be a fairly straightforward application of existing halfspace learning algorithms; the main contribution here is realizing that the existing online learners are enough.*

A: We view simplicity as an advantage of our work. Further, given that many works have studied the problem and that the online learning results have existed for a couple of decades, we believe that it is not straightforward.

*Q: There are a number of other papers giving adversarially robust learning guarantees for halfspaces, including (non-agnostic) learning for $L_p$ perturbations with random classification noise, and semi-agnostic learning for $L_2$.*

A: As the reviewer points out, this is an active research area. Our results complement the existing works mentioned, which only apply to more restricted noise models. As such, we do not view this point as a weakness of our work.

**Review #4**

*Q: Nevertheless, given the amount of work on closely related problems, this work is in some ways a little incremental. Also, one might argue that a stochastic noise model is often of more relevance in practice than the highly pessimistic agnostic noise model, where much stronger guarantees are generally possible for stochastic noise. (I feel that the agnostic model is still worthy of study and more relevant for some scenarios, though.)*

A: While the stochastic noise model might be more realistic, the existing works often assume random classification noise where each label is flipped independently with probably *exactly* $\eta < 0.5$. Known algorithms in this model do not extend naturally even to the case where the flip probability is *at most* $\eta$ (aka Massart Noise); such a limitation calls their practicality into question. On the other hand, our algorithms work in the most general agnostic noise model, which can be applied without strong assumptions about the specific random process creating the noise.

[Meta-Review · NeurIPS 2020]

The four reviewers in their initial reviews all recommended accepting. On the positive side, the topic is timely and the paper introduces techniques that might be of interest to other researchers in the field. On the negative side, the achieved results feel somewhat incremental considering recent related work. After the author reply and discussion, the reviewers are even more strongly unanimous about accepting the paper.